# Evidence that Mediator is essential for Pol II transcription, but is not a required component of the preinitiation complex in vivo

**Natalia Petrenko[1†], Yi Jin[1†], Koon Ho Wong[2†], Kevin Struhl[1*]**

[1]Department of Biological Chemistry and Molecular Pharmacology, Harvard Medical School Boston, Boston, United States; [2]Faculty of Health Sciences, University of Macau, Macau, China

**Abstract** The Mediator complex has been described as a general transcription factor, but it is unclear if it is essential for Pol II transcription and/or is a required component of the preinitiation complex (PIC) in vivo. Here, we show that depletion of individual subunits, even those essential for cell growth, causes a general but only modest decrease in transcription. In contrast, simultaneous depletion of all Mediator modules causes a drastic decrease in transcription. Depletion of head or middle subunits, but not tail subunits, causes a downstream shift in the Pol II occupancy profile, suggesting that Mediator at the core promoter inhibits promoter escape. Interestingly, a functional PIC and Pol II transcription can occur when Mediator is not detected at core promoters. These results provide strong evidence that Mediator is essential for Pol II transcription and stimulates PIC formation, but it is not a required component of the PIC in vivo.

*For correspondence: kevin@hms.harvard.edu

[†]These authors contributed equally to this work

## Introduction

Mediator is a highly conserved, transcriptional co-activator complex that physically bridges activator proteins bound at enhancers and Pol II bound at the promoter (*Allen and Taatjes, 2015*; *Plaschka et al., 2015*; *Robinson et al., 2015*; *Jeronimo et al., 2016*; *Petrenko et al., 2016*). In yeast, Mediator is recruited efficiently to enhancers by many activator proteins that mediate diverse stress responses (*Bhoite et al., 2001*; *Bryant and Ptashne, 2003*; *Kuras et al., 2003*; *Fan et al., 2006*), but poorly by activators that control ribosomal protein or glycolytic genes under optimal growth conditions (*Fan et al., 2006*; *Grünberg et al., 2016*). In addition, Mediator directly interacts with Pol II, and it can stimulate preinitiation complex (PIC) assembly, phosphorylation of the Pol II C-terminal domain (CTD) by TFIIH, and basal transcription in vitro (*Thompson et al., 1993*; *Kim et al., 1994*; *Guidi et al., 2004*; *Takagi and Kornberg, 2006*; *Esnault et al., 2008*; *Malik et al., 2017*).

In yeast cells, the PIC has been defined experimentally as the entity that contains Mediator and general transcription factors bound to the core promoter in vivo (*Wong et al., 2014*). However, the PIC is short-lived (estimated as 1/8 s by *Wong et al., 2014*), because Mediator only transiently associates with the core promoter; it rapidly dissociates from the PIC upon TFIIH-mediated phosphorylation of the Pol II CTD (*Jeronimo and Robert, 2014*; *Wong et al., 2014*). Such TFIIH-dependent dissociation of Mediator is important for efficient escape of Pol II from the promoter into the elongation phase of transcription (*Wong et al., 2014*). Upon Mediator dissociation and promoter escape of Pol II, the other general transcription factors remain at the core promoter as a post-escape complex (*Wong et al., 2014*).

Mediator consists of 25 subunits in *S. cerevisiae* (*Bourbon et al., 2004*) that are organized in 4 modules: the head, middle, tail and kinase modules (*Allen and Taatjes, 2015*; *Plaschka et al., 2015*; *Robinson et al., 2015*). The head module interacts with Pol II, and many head and some middle subunits are essential for yeast cell growth. The tail module directly contacts transcriptional activators, and loss of one or more tail subunits impairs but does not eliminate cell growth, although activator-dependent transcription can be affected (*Zhang et al., 2004*; *Ansari et al., 2012*; *Paul et al., 2015*). The kinase module has modest negative or positive effects on selected genes, and cells lacking this module grow well under many conditions (*Nemet et al., 2014*). During transcriptional activation, Mediator undergoes a compositional change in which the kinase module dissociates from the remainder of the complex upon interaction with Pol II; however, this dissociation is not rate-limiting for transcription (*Jeronimo et al., 2016*; *Petrenko et al., 2016*).

Mediator is required for yeast cell growth and is often considered to be a general transcription factor like TBP, TFIIB, and Pol II itself (*Thompson and Young, 1995*; *Takagi and Kornberg, 2006*). However, the issue of whether Mediator is essential for Pol II transcription in vivo is controversial and unresolved, largely because it has been addressed almost exclusively with a temperature-sensitive (ts) mutation of the essential subunit Med17(Srb4).

Loss of Med17 function at the restrictive temperature reduces mRNA levels to the same extent as observed for a ts mutation of Pol II, suggesting that Mediator is essential for Pol II transcription in vivo (*Thompson and Young, 1995*; *Holstege et al., 1998*). However, mRNA measurements are complicated by effects on mRNA stability and hence do not directly assess Pol II transcription. Inactivation of Med17 causes decreased TBP occupancy at all promoters tested (*Kuras and Struhl, 1999*; *Li et al., 1999*), and genome-wide experiments show a general decrease in Pol II occupancy (*Paul et al., 2015*) and nascent transcription (*Plaschka et al., 2015*). However, our reanalysis of data from (*Paul et al., 2015*) reveals that this general decrease in Pol II occupancy is only 2–3 fold (see Results), suggesting that there is substantial transcription in the absence of Med17 function. Consistent with this observation, in the *med17*-ts strain, transcription mediated by artificial recruitment of general transcription factors is reduced about 3-fold (*Lacombe et al., 2013*), and some genes can be activated (*Lee and Lis, 1998*; *McNeil et al., 1998*; *Li et al., 1999*). In addition, the *med17*-ts strains can grow at elevated temperatures in the presence of suppressor mutations in NC2, a TBP-interacting complex (*Gadbois et al., 1997*). In the *med17*-ts strain, the head domain of Mediator breaks up, but the tail module is still recruited to genes (*Linder et al., 2006*; *Paul et al., 2015*) and might contribute to the transcriptional function of Mediator.

Here, we comprehensively address whether Mediator is required for Pol II transcription by using the anchor-away system (*Haruki et al., 2008*) to rapidly deplete individual or multiple Mediator subunits from the nucleus. Our results provide evidence that Mediator is essential for Pol II transcription in vivo, but that Mediator modules that associate either with the enhancer or with the core promoter confer partial transcriptional activity. In addition, the results indicate that Mediator inhibits promoter escape, but is not an obligate component of the preinitiation complex.

## Results

### Substantial transcription persists upon depletion of essential Mediator subunits

Classic loss-of-function experiments to elucidate the function of genes essential for cell growth are always compromised by the inability to completely remove or inactivate the encoded gene product. As a consequence, various approaches have been used to reduce the function of essential gene products, such as ts mutants (*Horowitz and Leupold, 1951*; *Edgar and Lielausis, 1964*; *Hartwell, 1967*), inducible protein degradation via degron-tagged proteins (*Dohmen et al., 1994*; *Moqtaderi et al., 1996*; *Nishimura et al., 2009*), specific chemical inhibitors (*Bishop et al., 2000*), and anchor-away (*Haruki et al., 2008*). These approaches are complementary, and each of them has advantages and disadvantages. The anchor-away method permits the rapid removal of proteins from the nucleus under conditions where cells are not stressed by heat shock or other environmental insults (*Haruki et al., 2008*). Although cells are treated with rapamycin to induce the anchor-away process, the strains carry the *tor1*-1 mutation that blocks the physiological effects of rapamycin.

Most importantly, control anchor-away strains show comparable Pol II occupancy profiles in the presence or absence of rapamycin (*Wong et al., 2014*).

For a comprehensive analysis, we generated anchor-away strains for every essential Mediator subunit, and several non-essential subunits, and examined Pol II occupancy upon rapamycin treatment. Levels of the tagged Mediator subunits associated with the genome were reduced to background levels upon rapamycin addition (*Petrenko et al., 2016*); *Figure 1A*), indicating that the anchor-away procedure is efficient. Depletion of each Mediator subunit tested, including those essential for cell growth, does not lead to a global shutdown of Pol II transcription, but rather a modest decrease on average (*Figure 1B*). In contrast, depletion of TBP or Pol II leads to a drastic decrease in transcription (*Figure 1B*). For Mediator-depleted strains, the strongest decreases in Pol II occupancy are observed upon depletion of the essential head subunit Med17, the essential scaffold subunit Med14, and the essential middle subunit Med7. Depletion of Cdk8, the catalytic subunit of the kinase module, has very modest effects on Pol II occupancy (*Figure 1B*).

In the above experiments, genes are expressed at steady-state levels prior to depletion of the Mediator subunit. To address the effect of Mediator depletion on inducible transcription, we depleted cells of Med17 and then analyzed the rapid transcriptional activation response to heat shock and copper. In accord with modest transcriptional effects described above, heat shock induction of *HSP82* and copper induction of *CUP1* is reduced 2-fold in Med17-depleted cells (*Figure 1C*). This observation is consistent with previous observations of heat shock and copper induction in the *med17*-ts strain (*Lee and Lis, 1998*; *McNeil et al., 1998*; *Li et al., 1999*).

Our reanalysis of published genome-scale Pol II occupancy data in a *med17*-ts strain (*Paul et al., 2015*) reveals similar results to those obtained here with the Med17-depletion strain (*Figure 1—figure supplements 1A* and *2*); substantial transcription, albeit at an average 3-fold lower level upon loss of Med17 function. Quantitative analysis on ten additional genes confirms that the effect on Pol II occupancy when Med17 is depleted via anchor-away (*Figure 1C* and *Figure 1—figure supplement 3A*) is similar to that seen when Med17 is inactivated via the temperature-sensitive mutation (*Figure 1D* and *Figure 1—figure supplement 3B*). In both situations, loss of Med17 function leads to dissociation of other head and middle subunits from the enhancer, whereas the tail module remains (*Linder et al., 2006*; *Paul et al., 2015*; *Petrenko et al., 2016*; *Figure 1—figure supplement 1B*). More generally and as discussed below, the stronger effect on SAGA-dependent genes observed under conditions of Med17 depletion also occurs in the *med17*-ts strain (*Paul et al., 2015*). Thus, inactivation or depletion of Med17 by independent methods yields similar disruption of Mediator structure and quantitatively modest transcriptional effects.

## Mediator preferentially affects SAGA-dependent vs. TFIID-dependent genes

Yeast promoters have been mechanistically characterized as 'constitutive' or 'regulated' based on the presence of canonical TATA elements, poly(dA:dT) sequences, transcriptional activator binding sites, chromatin structure, and TFIID dependence (*Struhl, 1986*; *Chen et al., 1987*; *Struhl, 1987*; *Iyer and Struhl, 1995*; *Moqtaderi et al., 1996*). Genome-scale analysis has classified genes as TFIID- or SAGA-dependent based on the co-activators most important for expression (*Basehoar et al., 2004*; *Huisinga and Pugh, 2004*). In general, 'constitutive' genes are TFIID-dependent and 'regulated' genes are SAGA-dependent, although there is not a strict correspondence among individual genes.

For all head, middle, and tail subunits tested, SAGA-dependent genes are more strongly affected by Mediator depletion than TFIID-dependent genes (*Figure 2A* and *Figure 2—figure supplement 1*). The transcriptional profiles in these Mediator-depletion strains are similar, though not identical (*Figure 2B*). The relative importance of Mediator at SAGA-dependent vs. TFIID-dependent genes has been described in strains lacking the tail module (*Ansari et al., 2012*; *Paul et al., 2015*), but our results confirm those on other Mediator subunits that were reported while this work was in progress (*Jeronimo et al., 2016*). The relative importance of Mediator for SAGA-dependent vs. TFIID-dependent genes is also observed for Kin28, the kinase subunit of TFIIH (*Wong et al., 2014*). In contrast, depletion of Cdk8 kinase has only a minor effect on Pol II occupancy, with a distinct transcriptional profile that does not discriminate between SAGA- and TFIID-dependent genes.

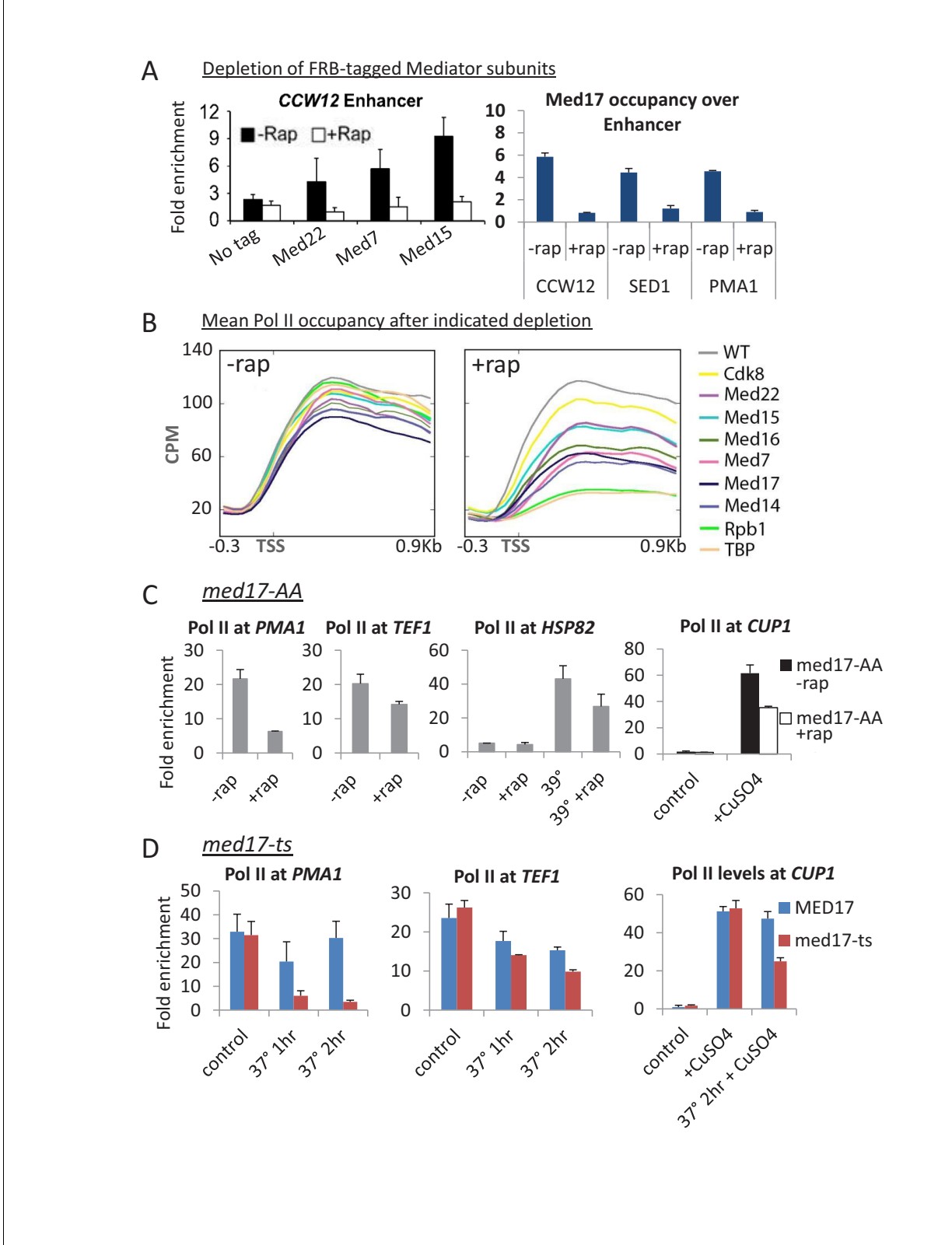

**Figure 1.** Substantial transcription persists upon anchor-away of essential Mediator subunits. (**A**) Occupancy of the indicated 3x-HA-FRB tagged Mediator subunit at the indicated enhancers prior to (-Rap) or after (+Rap) being depleted by anchor away. The parental strain containing no FRB-tagged protein was used as a negative control. The HA antibody was used except for Med17, in which case an antibody to the native protein was used. Data from (*Petrenko et al., 2016*). (**B**) Mean Pol II occupancy over ~400 transcribed genes prior to and after anchor-away of the indicated Mediator

*Figure 1 continued on next page*

*Figure 1 continued*

subunits, TBP, Pol II (Rpb1) and the parental strain (WT). Sequence reads were normalized as counts per million (CPM), and the curves were aligned relative to the transcription start site (TSS). (C) Pol II occupancy at the indicated constitutive and induced (by heat shock at 39°C or addition of copper) genes prior to and after Med17 anchor-away. (D) Pol II occupancy at constitutive genes and at the copper-inducible *CUP1* gene prior to and after heat inactivation of a *med17-ts* allele. An isogenic *MED17* strain was used as the control.

The following figure supplements are available for figure 1:

**Figure supplement 1.** Depletion of Med17 function via the ts mutant or anchor away results in similar transcriptional effects.

**Figure supplement 2.** Screenshots of Pol II occupancy for the indicated genes for samples analyzed in *Figure 1—figure supplement 1A*.

**Figure supplement 3.** Pol II occupancy at the indicated constitutive genes in the (A) Med17-AA (± rapamycin) and (B) *med17*-ts (at permissive temperature and 37°C for 1 or 2 hr) strains.

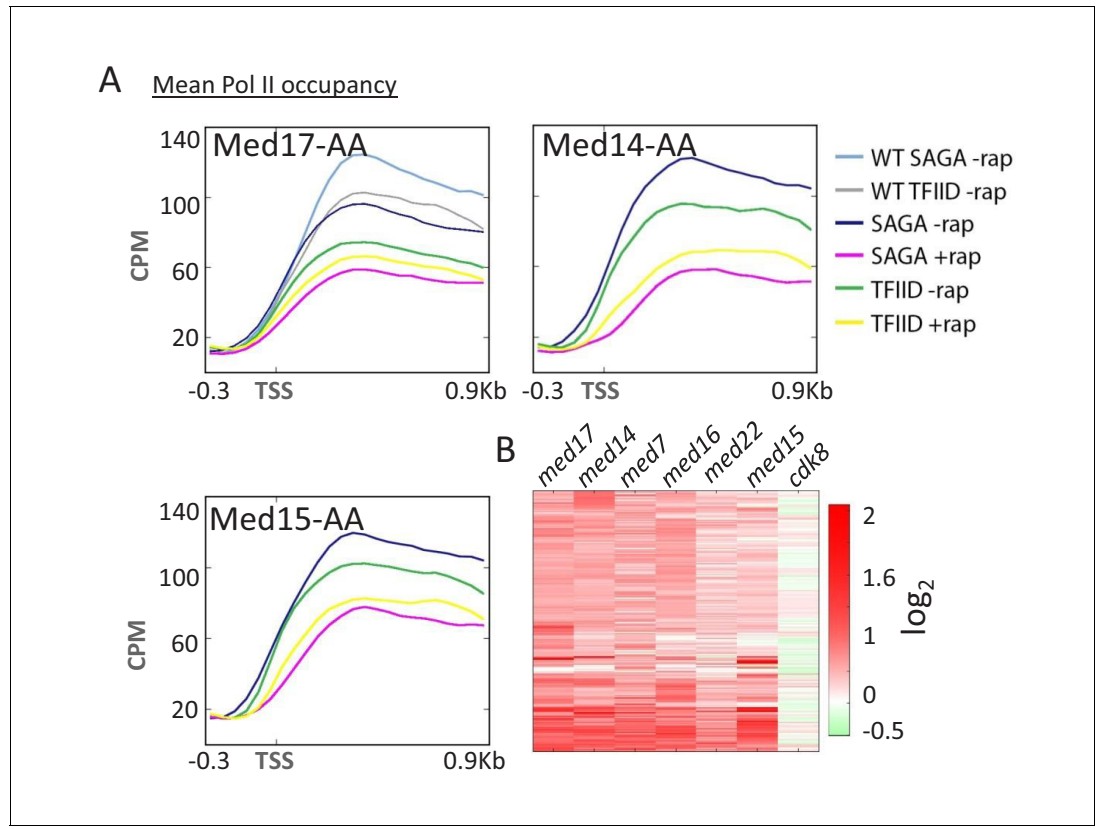

**Figure 2.** Mediator preferentially affects SAGA-dependent vs. TFIID-dependent genes. (A) Mean Pol II occupancy for the indicated groups of genes (SAGA or TFIID-dependent) prior to and after anchor-away of Med17, Med14, and Med15, as well as for the parent strain (WT). Sequence reads were normalized as counts per million (CPM), and the curves were aligned relative to the transcription start site (TSS). (B) Heat map of Pol II levels (log₂ scale) at individual genes before vs. after anchor-away of the indicated Mediator subunits. Occupancy changes upon anchor-away are indicated as decreases (red) or increases (green).

The following figure supplement is available for figure 2:

**Figure supplement 1.** Mean Pol II occupancy for the indicated groups of genes (SAGA or TFIID-dependent) prior to and after anchor-away of Sin4 (Med16), Srb6 (Med22), Srb10 (Cdk8), and Med7.

## Depletion of Mediator causes a downstream shift in the Pol II profile, indicating that Mediator inhibits promoter escape

Kin28-dependent phosphorylation of the Pol II CTD causes dissociation of Mediator from the PIC (*Jeronimo and Robert, 2014*; *Wong et al., 2014*), which is important for efficient escape of Pol II from the promoter (*Wong et al., 2014*). In particular, depletion of Kin28 causes increased Mediator occupancy at the core promoter (*Jeronimo and Robert, 2014*; *Wong et al., 2014*) and an upstream shift in the Pol II profile (*Wong et al., 2014*), indicative of a defect in promoter escape. Conversely, depletion of Mediator head or middle subunits causes a downstream shift in the Pol II profile (*Figure 3A*). This downstream shift is not observed upon depletion of Mediator subunits in the tail or kinase module (*Figure 3B*). In addition, the Pol II profile is unaffected under conditions of TBP depletion, even though Pol II transcription is drastically reduced (*Figure 3—figure supplement 1*). These observations provide complementary evidence that Mediator, via the head and middle subunits, inhibits promoter escape in vivo. In addition, this downstream shift in the Pol II profile upon

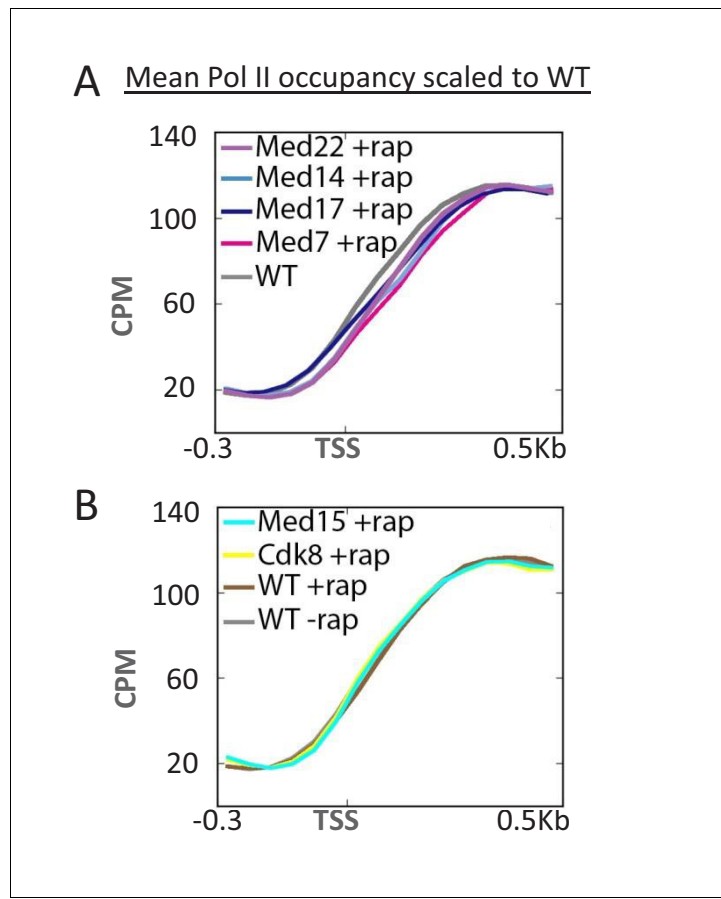

**Figure 3.** Depletion of Mediator head and middle subunits causes a downstream shift in the Pol II profile. (**A**) Overlaid mean Pol II occupancy curves scaled to 100% (maximum levels) after anchor-away of the indicated head and middle subunits of Mediator. (**B**) Overlaid mean Pol II occupancy curves scaled to 100% after anchor-away of the indicated tail and kinase module subunits, as well as for the parent strain (WT) before and after rapamycin addition. Sequence reads were normalized as counts per million (CPM), and the curves were aligned relative to the transcription start site (TSS).

The following figure supplement is available for figure 3:

**Figure supplement 1.** Overlaid mean Pol II occupancy curves scaled to 100% after anchor-away of TBP, as well as for the parent strain (WT) before and after rapamycin addition.

Mediator depletion provides evidence that Pol II transcription can occur even when Mediator is not present at the PIC (see Discussion).

## Pol II transcription can occur from preinitiation complexes lacking Mediator

Pol II transcription occurs when Mediator subunit occupancy is not detected (*Figure 1A,B*), and depletion of Mediator alters the Pol II profile (*Figure 3*), suggesting that Mediator is not an essential component of the PIC. However, as Mediator occupancy was only assessed at enhancers due to its transient interaction with core promoters, it remained formally possible that sufficient Mediator was associated at core promoters to permit a modest level of transcription. To address this possibility, we utilized the fact that Mediator association with core promoters is stabilized and can be assessed under conditions where Kin28 is depleted or inactivated (*Jeronimo and Robert, 2014*; *Wong et al., 2014*).

When Med17 and Kin28 are depleted simultaneously, the level of Pol II occupancy in the coding region is roughly comparable to that observed when these proteins are depleted individually (*Figure 4A* and *Figure 4—figure supplement 1*). However, for all cases tested, Mediator occupancy (Med8 and Med22 subunits) at the core promoter is greatly reduced upon simultaneous depletion of Med17 and Kin28 as compared with depletion of Kin28 alone (*Figure 4A* and *Figure 4—figure supplements 1* and *2*). This observation is true for genes that are continuously transcribed (*CCW12, TEF1, PMA1*), as well as those induced after depletion by heat shock (*HSP12, HSP82, SED1*) or copper (*CUP1*). Most importantly, the Mediator:Pol II occupancy ratio at all these genes upon simultaneous depletion of Med17 and Kin28 is far below the consistent ratio observed in Kin28 depletion strains (*Jeronimo and Robert, 2014*; *Wong et al., 2014*). In all cases tested, TBP and TFIIB occupancy at the core promoters is in excellent accord with Pol II occupancy in the coding regions (*Figure 4B* and *Figure 4—figure supplements 1* and *2*), indicative of a functional PIC.

In contrast to these results, simultaneous depletion of Kin28 and TBP drastically reduces transcription and TBP, TFIIB, and Mediator occupancies at the core promoter (*Figure 5A*). Moreover, under conditions where TBP/Kin28 depletion is less efficient (obtained by reducing the rapamycin concentration by a factor of four), the level of transcription is reduced in accord with the reduction in TBP, TFIIB, and Mediator occupancy (*Figure 5B*). Thus, in the absence of Kin28, depletion of TBP affects the level but not the composition (i.e. the relative occupancy of the components) of the PIC, whereas depletion of Med17 alters the composition of a transcriptionally-competent PIC. These observations suggest that Pol II transcription and hence a functional PIC can occur in the absence of Mediator at the core promoter, and hence that Mediator is not an essential component of the PIC in vivo (see Discussion).

## Mediator is important, but not essential, for serine 5 phosphorylation of the Pol II C-terminal domain

Mediator can stimulate Kin28-dependent phosphorylation of the Pol II CTD at serine 5 residues in vitro (*Guidi et al., 2004*; *Esnault et al., 2008*; *Nozawa et al., 2017*), but this activity has never been examined in vivo. In accord with the biochemical observations, depletion of all Mediator subunits tested causes decreased phosphorylation of serine 5 residues in the Pol II CTD (normalized to Pol II levels) at all core promoter regions examined (*Figure 6*). However, the level of CTD-serine 5 phosphorylation upon Mediator depletion is higher than observed when Kin28 is depleted. As expected (*Komarnitsky et al., 2000*), CTD-serine 5 phosphorylation levels were low near the 3' end in all strains (*Figure 6—figure supplement 1*). Thus, Mediator contributes to, but is not fully responsible for, CTD-serine 5 phosphorylation in vivo.

## Pol II transcription is virtually eliminated when Mediator head, middle, and tail modules are simultaneously inactivated

Although considerable transcription persists when Med17 or essential Mediator subunits are depleted, the tail module still associates with enhancers and might influence transcription. To address whether transcription can be abolished when all Mediator modules are depleted or eliminated, we generated a *med17*-AA derivative lacking the genes encoding the Med3 and Med15 tail

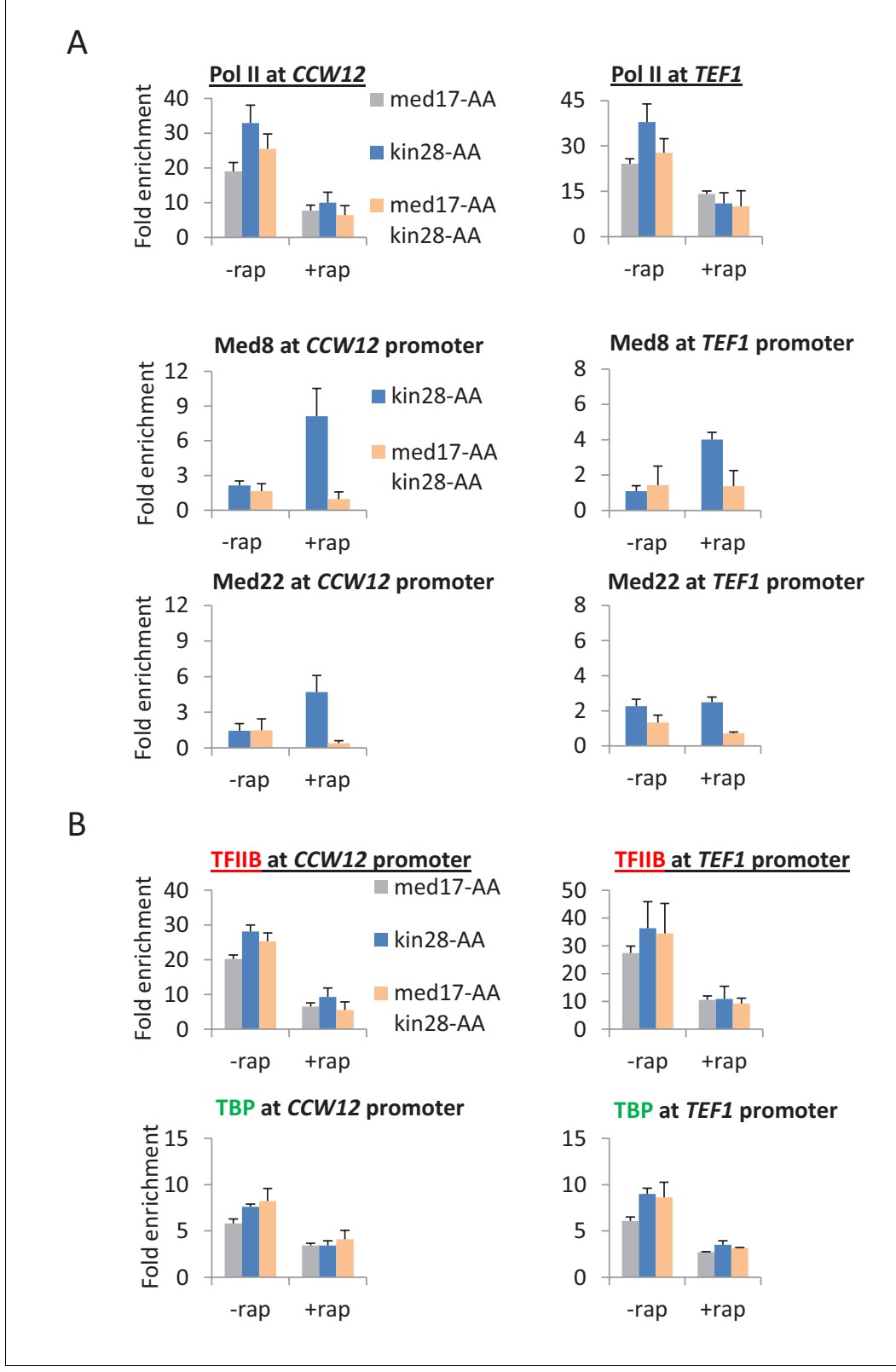

**Figure 4.** Pol II transcription can occur from preinitiation complexes lacking Mediator. (**A**) Pol II occupancy in the coding regions and Mediator subunits at the promoters of the *CCW12* and *TEF1* genes, before or after anchor-away of Med17, Kin28, or both simultaneously. (**B**) TBP and TFIIB occupancy at the *CCW12* and *TEF1* promoters in the same samples.

*Figure 4 continued on next page*

*Figure 4 continued*

The following figure supplements are available for figure 4:

**Figure supplement 1.** Pol II transcription can occur from preinitiation complexes lacking Mediator.

**Figure supplement 2.** Pol II transcription can occur from preinitiation complexes lacking Mediator.

subunits. In strains lacking Med3 and Med15, a third tail subunit, Med2, is no longer recruited to genes (*Zhang et al., 2004*; *Paul et al., 2015*).

In the absence of rapamycin, Pol II levels at *PMA1*, *CCW12*, and *TEF1* in the triple mutant strain are reduced compared to the wild-type (*Figure 7A*). Depleting Med17 in this strain causes a further decrease in transcription of all three genes, close to or at the background level of detection (*Figure 7A*). The triple depletion strain can induce *SSA4* and *HSP82* upon heat shock or *CUP1* in response to copper, albeit at a much lower level than wild-type cells or cells deleted either for the tail module (*Figure 6A*) or Med17 (*Figure 1C*). Comparably low levels of transcription are observed in a strain where Med3, Med15, and Med17 were simultaneously depleted via anchor-away (*Figure 7B*). In contrast, simultaneous depletion of the essential subunits Med22 (head module) and Med7 (middle module) results in substantial levels of transcription, comparable to that observed upon Med17 depletion (*Figure 7B*). Thus, depletion of all three Mediator modules has a stronger transcriptional effect than conditions where the tail module is present at enhancers (Med17 depletion) or the head and middle modules are present at core promoters (deletion of tail subunits).

The weak heat shock response observed in the triple depletion strain could represent either a low level of Mediator-independent transcription or incomplete depletion of Mediator subunits. As it is impossible to directly exclude the possibility of incomplete depletion, we examined heat shock and copper induction in strains depleted of Pol II or TBP by the same anchor-away method (*Figure 7C*). We presume that any transcription observed in TBP- or Pol II-depleted strains represents incomplete depletion. In both cases, there is a very low level of transcriptional activation, roughly comparable to (although perhaps slightly lower than) that occurring in the triple depletion strain. More generally, genome-scale, RNA-seq analysis indicates that the level of Pol II transcription upon depletion of all Mediator modules is indistinguishable from that occurring upon TBP depletion (*Figure 7D*). These observations indicate that most, and perhaps all, of the weak activation in the triple depletion strain reflects incomplete depletion of Mediator subunits.

## Growth of Mediator-depletion strains

For all 18 Mediator subunits tested, depletion of any individual subunit (or the combination of the essential subunits Med22 and Med7) does not prevent cells from growing at 30°C (*Figure 8A*). As deletion of some Mediator subunits prevents cell growth, these observations indicate that depletion of Mediator subunits by anchor-away is incomplete. Interestingly, as seen in the *med17*-ts strain, the Med17 and many other anchor-away strains are unable to grow at 37°C (*Figure 8B*). As the *med17*-ts and Med17 depletion strains have effects on transcription (*Figure 1C,D*), the failure of the *med17*-ts strain to grow at elevated temperature may not be due to complete inactivation of Med17 but rather the requirement for higher levels of Mediator function to support growth under stressful conditions.

In striking contrast to all individual Mediator subunits tested, growth at 30°C is abolished upon depletion of individual subunits of any general transcription factor (TBP, TFIIA, TFIIB, TFIIE, TFIIF, TFIIH, Pol II) by the same anchor-away method (*Figure 8A*). However, simultaneous depletion/ removal of all Mediator modules (either by depleting Med17 in the tail deletion strain (*med3Δ med15Δ med17*-AA) or triple anchor-away depletion of the same subunits (*med3*-AA *med15*-AA *med17*-AA) results in extremely poor growth at 30°C (*Figure 8C*). Thus, depletion of all Mediator modules causes drastic effects on transcription and cell growth, whereas depletion of individual Mediator subunits has more modest effects. This striking dichotomy suggests that viability of Mediator-depletion strains is not due to incomplete depletion by the anchor-away method per se. Moreover, incomplete anchor-away-mediated depletion cannot easily explain why cells subject to

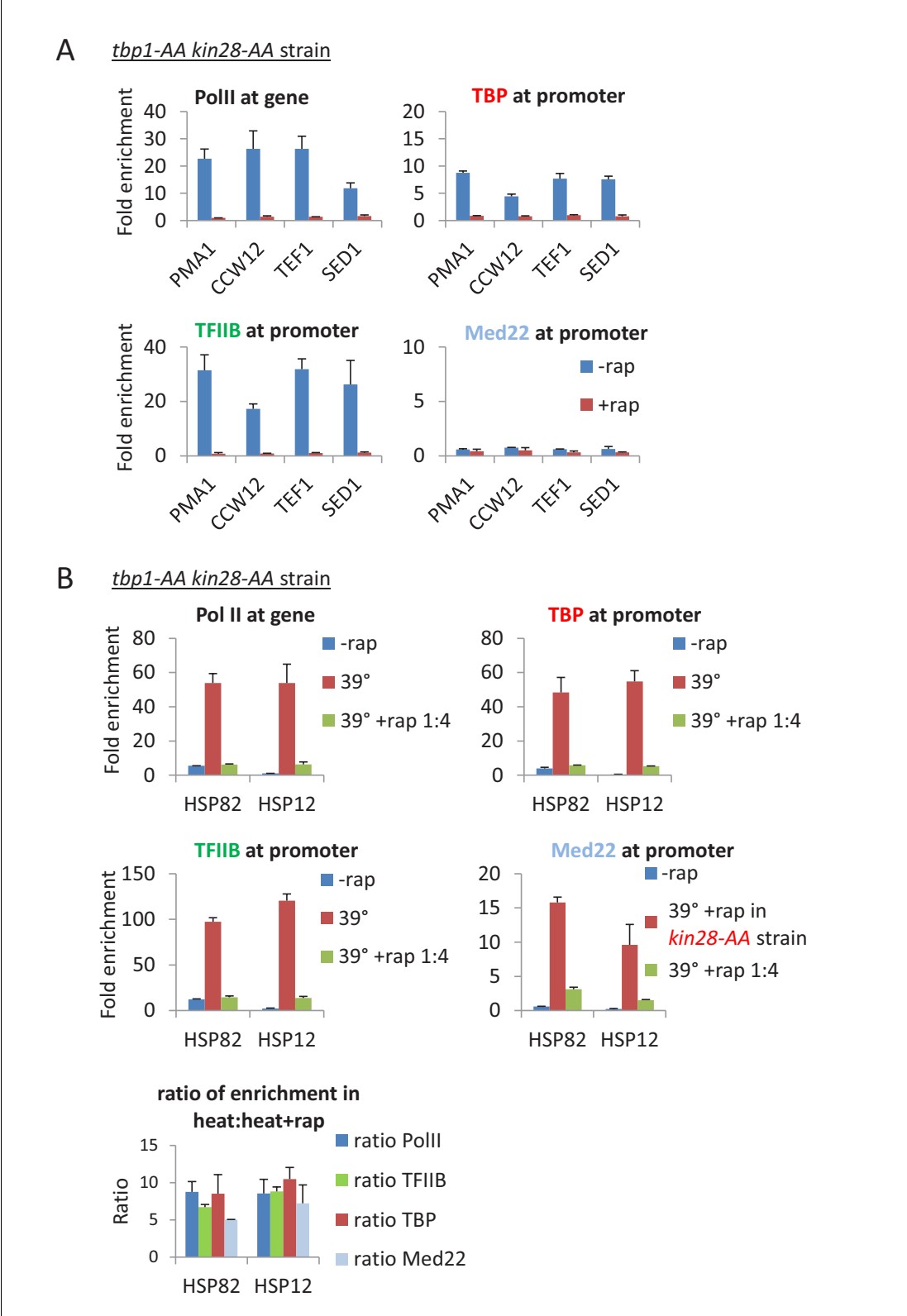

**Figure 5.** Depletion of TBP does not alter the composition of the preinitiation complex. (**A**) Occupancy of Pol II in the coding regions, and Med22, TBP, and TFIIB at the promoters of the indicated genes before or after simultaneous depletion of TBP and Kin28. (**B**) Occupancy of Pol II in the coding regions, and Med22, TBP, and TFIIB at the promoters of the indicated heat shock genes before or after a heat shock at 39°C in cells that were or were not incompletely depleted for TBP and Kin28 by using rapamycin at 25% of the usual concentration. As Mediator can only be detected at the core

*Figure 5 continued on next page*

*Figure 5 continued*

promoter upon Kin28 depletion, Med22 occupancy at heat shock promoters were tested in a *kin28*-AA strain. The relative occupancy ratios of Pol II, TBP, TFIIB, and Med22 in heat shocked cells that were or were not incompletely depleted is shown below.

simultaneous depletion of essential Mediator subunits in the head (Med22) and middle (Med7) module are viable, whereas cells simultaneously depleted of the essential Med17 and two non-essential tail subunits (Med3 and Med15) are inviable.

To explain why some Mediator subunits are essential, we suggest that transcription initiated from Mediator-lacking PICs is substantial but insufficient for cell growth. Incomplete depletion of an essential subunit allows enough additional transcription to put cells over the life/death threshold. This does not occur when transcription is virtually eliminated when all Mediator modules or individual general transcription factors are depleted. Consistent with the idea of a viability threshold, the viable Mediator-depletion strains grow more slowly than parental strains at 30°C and not at all at 37°C. The difference between life and death could be due to overall reduced (but not eliminated) transcription or reduced transcription of one or more genes. We also note that, unlike inducible degron-based methods (*Dohmen et al., 1994*; *Moqtaderi et al., 1996*; *Nishimura et al., 2009*), the anchor-away approach does not destroy the protein but rather anchors it to the ribosome. As such, it is formally possible that Mediator might have some non-chromosomal function, and in this regard Mediator has post-transcriptional roles (*Carlsten et al., 2013*; *Conaway and Conaway, 2013*; *Allen and Taatjes, 2015*).

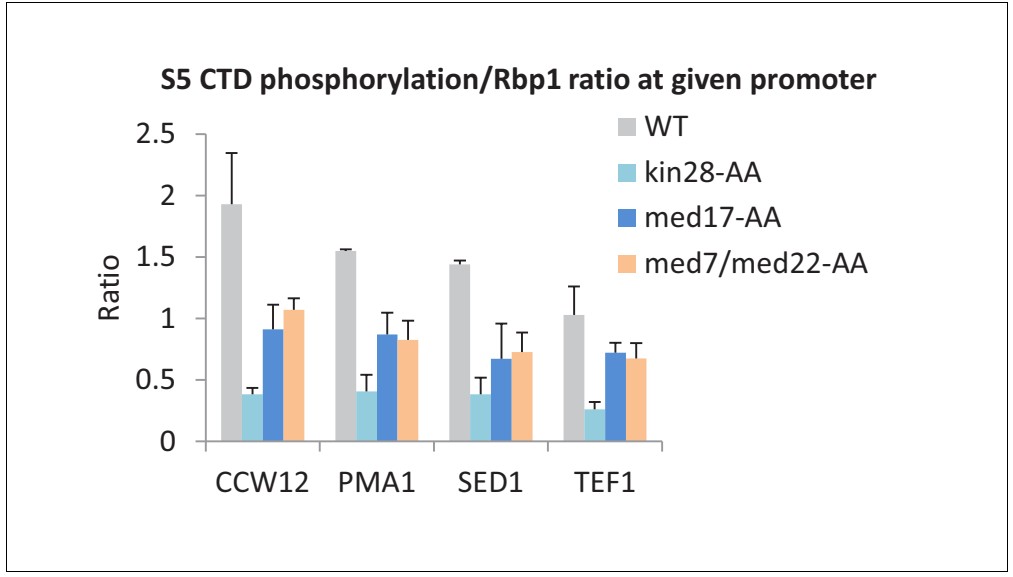

**Figure 6.** Mediator is important, but not essential, for serine 5 phosphorylation of the Pol II CTD. The ratio of S5-phosphorylated CTD levels relative to total Rpb1 levels at the promoters of the indicated genes, depletion of the indicated proteins or the control strain (WT).

The following figure supplement is available for figure 6:

**Figure supplement 1.** The ratio of S5-phosphorylated CTD levels relative to Rpb1 levels at the transcription stop sites of the indicated genes prior to or after Kin28 anchor-away, Med17 anchor-away, or a double anchor-away of Med7 and Med22, as well as in the parent strain (WT).

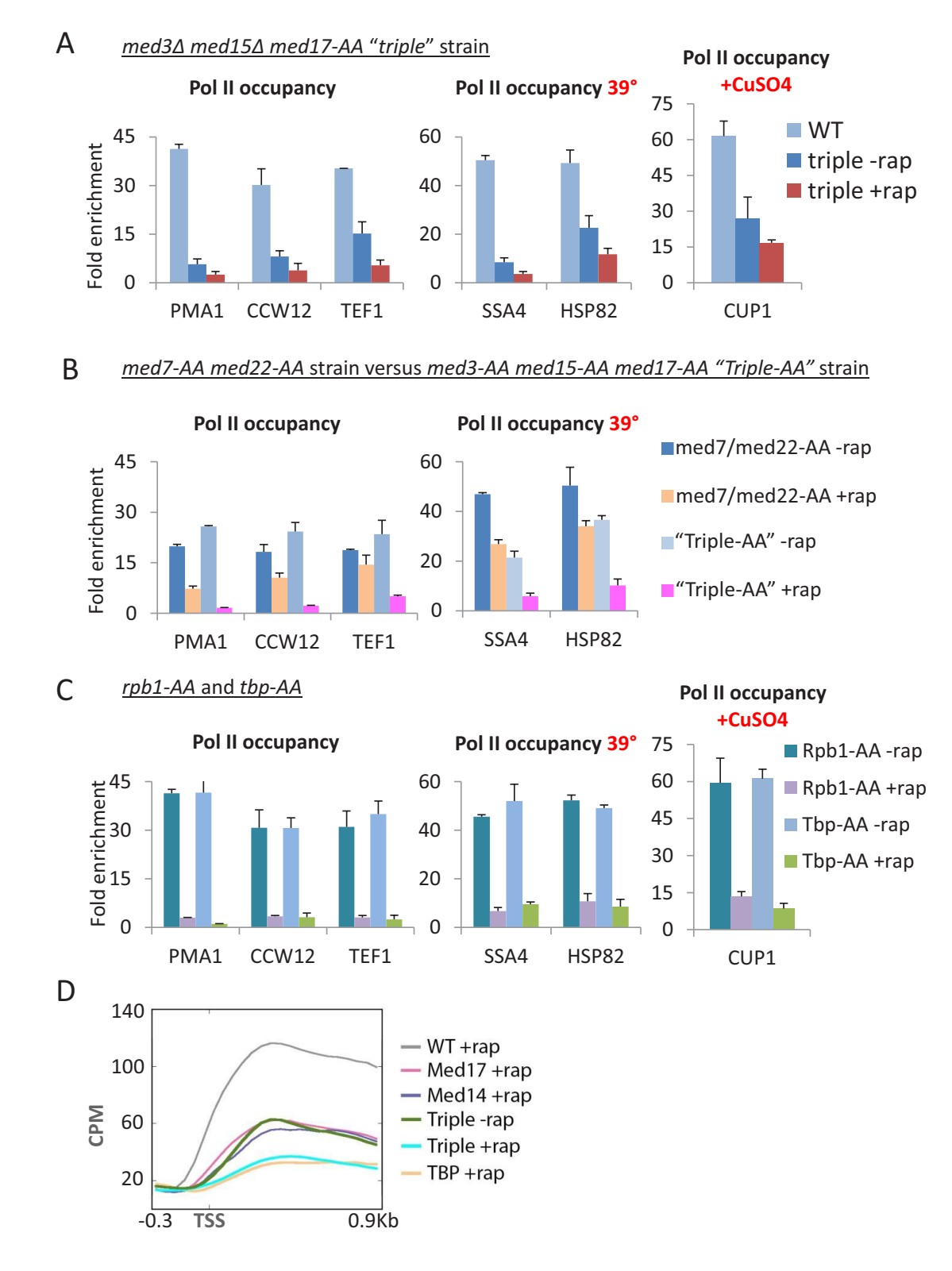

**Figure 7.** Pol II transcription is virtually eliminated when Mediator head, middle, and tail modules are simultaneously inactivated. (**A**) Pol II occupancy in the coding regions of the indicated constitutive and inducible genes for the parent strain (WT) and 'triple' mutant strain, in which the tail module is inactivated via deletion of *med3* and *med15*, and the head and middle modules are inactivated via anchor-away of Med17. (**B**) Pol II occupancy in the coding regions of the indicated constitutive and inducible genes for the parent strain (WT) and a strain simultaneously depleted for Med3, Med15 (tail

*Figure 7 continued on next page*

*Figure 7 continued*

module), and Med17 (head module) or for Med7 (middle module) and Med22 (head module). (**C**) Pol II occupancy in the same genes prior to and after anchor-away of Rpb1 and TBP. (**D**) Mean Pol II occupancy over ~400 transcribed genes in strains depleted for Med14 or Med17 as well as the triple mutant strain (before and after rapamycin) and the parental strain (WT). Sequence reads were normalized as counts per million (CPM), and the curves were aligned relative to the transcription start site (TSS).

## Discussion

### Evidence that Mediator is essential for Pol II transcription in vivo

Simultaneous depletion of subunits in the head, middle, and tail modules is the most stringent test of Mediator function in vivo. Under this condition (triple depletion strain), there is a drastic effect on Pol II occupancy at genes expressed at steady-state prior to depletion or induced after depletion. The magnitude of the transcriptional defect is roughly comparable to that observed upon depletion of TBP or Pol II by the same method. In addition, cells depleted for all Mediator modules grow extremely poorly, unlike the case for strains depleted for individual Mediator subunits. The transcriptional and growth effects upon depletion of all Mediator modules may be slightly less pronounced than upon depletion of TBP or Pol II. These subtle differences could be due to a very low level of Mediator-independent transcription, or they might simply reflect a very subtle difference in depletion efficiency, and hence be are an experimental artifact. Thus, while impossible to prove conclusively due to the inherent limitations of studying proteins that are essential for cell viability, our results provide strong evidence that Mediator is essential for Pol II transcription in vivo.

### Mediator modules make independent contributions to the overall transcriptional function of Mediator

Our results suggest that Mediator modules that associate either with the enhancer or with the core promoter confer partial transcriptional activity, and hence contribute independently to the overall transcriptional function. In this view, depletion/inactivation of Med17 (and other essential subunits) has a relatively modest transcriptional effect, because the tail module remains associated with the enhancer. The molecular basis of this tail-specific function is unknown, but it might reflect the ability of the tail module to interact with a component of the basic Pol II machinery and/or to increase the association of other co-activators (e.g. SAGA or Swi/Snf) with the enhancer. It is also possible that the tail module might increase recruitment of the very low levels of the head and middle subunits to the promoter, but not affect PIC function directly. Conversely, removal of the tail module also has a relatively modest effect on Pol II transcription, because the middle and head modules can associate with the PIC at the core promoter (*Jeronimo et al., 2016*; *Petrenko et al., 2016*). This explanation is consistent with the very low level of Mediator at enhancers that drive expression of ribosomal protein and glycolytic genes (*Fan et al., 2006*).

The independent functions of Mediator modules are consistent with the observation that SAGA-dependent genes are more affected than TFIID-dependent genes upon depletion of Mediator subunits. By virtue of TAF-DNA interactions (*Verrijzer et al., 1995*; *Oelgeschläger et al., 1996*; *Burke and Kadonaga, 1997*), TFIID strengthens the interaction of the basic Pol II machinery with the core promoter, thereby making the Mediator-dependent connection between enhancer and promoter less important at TFIID-dependent genes. In contrast, transcription of SAGA-dependent genes relies on TBP, not TFIID, and hence Mediator is needed to efficiently connect the enhancer and core promoter.

### Mediator is not an obligate component of the preinitiation complex in vivo

As is the case for general transcription factors, the entire Mediator complex is critical for Pol II transcription and hence PIC formation in vivo. However, Mediator interacts both with the enhancer and core promoter, and individual Mediator modules have independent effects on transcription. Furthermore, unlike general transcription factors, Mediator is not required for basal transcription and hence PIC formation in vitro. Thus, it is unclear whether Mediator, like general transcription factors, is an

**Figure 8.** Growth of Mediator depletion strains. (**A**) growth on YPD (2.5 days) in the presence or absence rapamycin upon anchor-away mediated depletion of the Mediator subunit (light blue shading) or general transcription factor (dark blue shading) corresponding to the key in the upper panel. WT corresponds to the parent strain. (**B**) Growth of the *med17*-anchor-away strain and of the parent strain on YPD at 30° or 37° in the presence or absence rapamycin. (**C**) Growth of the parental, 'triple' mutant, and triple depletion strain, on YPD at 30° in the presence or absence of rapamycin.

obligate component of the PIC in vivo. Two independent observations presented here strongly suggest that considerable transcription can occur from PICs that lack Mediator.

First, and most directly, cells depleted simultaneously for Med17 and Kin28 support substantial Pol II transcription even though a very low level of Mediator is detected at the core promoter. Furthermore, TBP and TFIIB occupancies in such cells are in accord with Pol II occupancy, indicative of a

functional PIC in the apparent absence of Mediator. In contrast, cells depleted only for Kin28 have a much higher level of Mediator at the core promoter, even though Pol II, TBP, and TFIIB occupancies are comparable. These low Mediator:Pol II, Mediator:TBP, and Mediator:TFIIB occupancy ratios at the core promoter upon simultaneous depletion of Med17 and Kin28 provides very strong evidence of a transcriptionally competent, Mediator-independent PIC. As discussed below, this conclusion does not rely on the degree of Mediator depletion per se, but rather direct observation at the core promoter.

Second, depletion of Mediator head or middle subunits causes a downstream shift in the Pol II occupancy profile. This non-wild-type Pol II profile is very difficult to explain by incomplete depletion per se (see below), and hence it strongly supports the idea of transcription initiated from a PIC lacking Mediator. Notably, this downstream shift in the Pol II profile is not observed upon depletion of tail subunits that do not directly interact with general transcription factors at the PIC, yet have comparable quantitative effects on Pol II occupancy. In addition to these two major arguments, our conclusion is supported by the dichotomy between Mediator and general transcription factors with respect to growth properties upon depletion.

Can incomplete depletion of Mediator explain the above observations that are the basis of our conclusion that Mediator is not an obligate component of the PIC? Mediator is not completely depleted in our experiments, and complete elimination of any essential protein is impossible. However, the definition of incomplete depletion is that the small amount of remaining protein is structurally and functionally identical to the protein prior to depletion. Thus, incomplete depletion of a general transcription factor will reduce (but not eliminate) transcription and its occupancy at the core promoter, but it will not affect either the relative ratios of general transcription factors at core promoters (i.e. PIC level) or the Pol II profile. Indeed, incomplete depletion of TBP not only reduces transcription, but it also reduces to comparable extents the levels of general transcription factors and Mediator at the core promoter. In contrast, depletion of Med17 drastically reduces Mediator occupancy at the core promoter, whereas it has only modest and comparable effects on occupancy of TBP, TFIIB, and Pol II. Thus, as incomplete depletion of Mediator cannot explain the key observations, our results indicate that (1) Mediator is not an obligate component of the PIC, (2) transcription can occur from a PIC lacking Mediator, and (3) that transcription from a Mediator-lacking PIC escapes the promoter more easily than a Mediator-containing PIC.

## Mechanistic implications

Mediator is essential for Pol II transcription, yet is not an obligate component of the PIC, and this apparent paradox cannot be explained by the classic Pol II recruitment model in which Mediator bridges the enhancer (via the tail domain) and core promoter (via the head domain). One possibility is that Mediator performs a catalytic function at the core promoter that alters the activity of the PIC in a manner that is essential for transcription. Except for a kinase subunit (Cdk8 in yeast) that has minimal effects on transcription, Mediator does not have any known enzymatic activities. However, Mediator can affect the conformation of Pol II (*Plaschka et al., 2015*; *Tsai et al., 2017*), so a Mediator-induced conformational effect could be a catalytic, yet essential function for Pol II transcription. In addition, biochemical experiments have suggested that Mediator functions as an assembly factor that facilitates PIC maturation through different stages (*Malik et al., 2017*).

Alternatively, the tail module that remains associated at the enhancer upon Med17 depletion could have an independent transcriptional function that does not involve its connection to the middle and head modules. For example, the tail module could interact directly with Pol II or a general transcription factor, thereby stimulating PIC formation. The tail module might indirectly stimulate PIC formation via a direct interaction with the SAGA co-activator complex, whose Spt3 subunit of SAGA interacts with TBP. This suggested mechanism would not only permit increased recruitment of a Mediator-lacking PIC, but it would also explain the observation that SAGA-dependent genes are more affected than TFIID-dependent genes upon depletion of Mediator subunits. These proposed mechanisms, and others not mentioned, can explain why the Mediator is essential for Pol II transcription even though it is not an obligate component of the PIC and hence is different from a general transcription factor.

## Materials and methods

### Yeast strains and growth conditions

Strains used in this study are listed in *Supplementary file 1*. Anchor-away strains were constructed as described previously (*Wong et al., 2014*), except for the Med17 strain which was kindly provided by Francois Robert. For spotting assays, yeast cells were grown to an $OD_{600}$nm of 0.3–0.5, diluted to 0.1, and 5-fold serial dilutions of cells were spotted on YPD medium with or without 1 µg/ml rapamycin; the plates were kept at 30°C or 37°C for 48–60 hr. For anchor-away, strains were grown in SC liquid media to an $OD_{600}$nm of 0.4, and rapamycin was then added to a final concentration of 1 µg/ml. For 39°C heat shock, cells (pretreated or not with rapamycin for 45 min) were grown at 30°C, the culture was filtered, transferred to pre-warmed 39°C media, and grown at 39°C for 15 min in the presence or absence of rapamycin. For 37°C heat inactivation of the *med17*-ts strain, a similar procedure was followed, but with media pre-warmed to 37°C, and cells were grown at 37°C for 1 or 2 hr. For *CUP1* induction with $CuSO_4$, 1 mM $CuSO_4$ (final concentration) was added for 15 min.

### Chromatin immunoprecipitation (ChIP)

Chromatin, prepared as described previously (*Fan et al., 2008*), from 5 ml of cells ($OD_{600}$nm ~0.5) was immunoprecipitated with antibodies against Pol II unphosphorylated CTD (8WG16, Covance), CTD-phosphorylated on serine 5 (3E8, Millipore), c-Myc (9E10, Santa Cruz), HA (F-7, Santa Cruz), TBP (a kind gift from Steve Buratowski), TFIIB, or Med17 (a kind gift from Steve Hahn). Immunoprecipitated and input samples were analyzed by quantitative PCR in real time using primers for genomic regions of interest and a control region from chromosome V to generate IP:input ratios for each region. The level of protein association to a given genomic region was expressed as fold-enrichment over the control region. For qPCR analysis, 3 to 4 biological replicates were performed for each experiment (biological replicates were culture samples collected on separate days, with lysis and IP performed on separate days). During qPCR analysis, each sample was tested in triplicate (3 'technical replicates') to avoid qPCR error, and the triplicates were averaged. If one of the triplicates differed by more than 2-fold from the other two, it was discarded as an outlier. Error bars represent the standard deviation between the 3 or 4 biological replicates.

### ChIP-seq and data analyses

Barcoded sequencing libraries from ChIP DNA (two biological replicates per strain) were constructed as described previously (*Wong et al., 2013*). Sequence reads were mapped using Bowtie available through the Galaxy server (Penn State) with the following options: *-n 2, -e 70, -l 28, -v -1, -k 1, -m -1*. Pol II occupancy of a gene was calculated by summing the number of ChIP-seq reads within an appropriate region, normalized to the respective surveyed window size, and is expressed as counts per million mapped reads (CPM). Normalization was also performed with respect to the median Pol II levels at the silent loci (*HML* and *HMR*) and a non-transcribed region of Chromosome V set as the 'background' level. Pol II occupancy peaks were called using MACS available through the Galaxy server (Penn State) with tag size set to 35, bandwidth to 150–300 bp, and the *P*-value cutoff at $1e^{-05}$. Mean occupancy curves were generated using Galaxy deepTools (Freiburg, Germany), scaled relative to the number of mapped reads and fragment size, and expressed as counts per million mapped reads (CPM). TFIID- and SAGA-dependent genes were defined previously (*Basehoar et al., 2004*; *Huisinga and Pugh, 2004*). Clustering was performed using the CIMminer (NCI) average linkage algorithm and Matlab. Pol II occupancy profiles for individual Mediator depletion conditions were generated by averaging the values from 2 replicates, defining 100% as the maximal value at +400 to +500 (which is comparable to levels further downstream), and then normalizing all values to the 100% value in that strain. The *p*-values comparing Mediator depletion to the wild-type control strain at position +100 downstream of the TSS are as follows: Med22 (0.0004); Med7 (0.00007); Med14 (0.015); Med17 (0.2). The overall significance is considerably higher because these *p*-values at +100 do not consider differences in Pol II occupancy at other positions, which are clearly apparent in *Figure 3*. The ChIP-sequencing data and associated files are available through the Gene Expression Omnibus (GEO) under the accession number GSE93190. For analysis of the Pol II occupancy data in *Paul et al. (2015)*, data was downloaded from the NCBI Sequence Read Archive under the accession number SRP047524.

## Acknowledgements

We thank Francois Robert for Med17 anchor-away strain, Steve Hahn for antibodies against Med17, Steve Buratowski for antibodies against TBP, and Răzvan Chereji for Matlab scripts. This work was supported by a Croucher Foundation Fellowship and research grants MYRG2015-00186-FHS and MYRG2016-0-0211-FHS from the University of Macau to KHW and by a research grant to KS from the National Institutes of Health (GM30186).

## Additional information

### Competing interests

KS: Reviewing editor, *eLife*. The other authors declare that no competing interests exist.

### Funding

| Funder | Grant reference number | Author |
|---|---|---|
| National Institutes of Health | GM 30186 | Kevin Struhl |
| Croucher Foundation | | Koon Ho Wong |
| University of Macau | MYRG2015-00186-FHS | Koon Ho Wong |
| University of Macau | MYRG2016-0-0211-FHS | Koon Ho Wong |

The funders had no role in study design, data collection and interpretation, or the decision to submit the work for publication.

### Author contributions

NP, Conceptualization, Data curation, Formal analysis, Validation, Investigation, Visualization, Methodology, Writing—original draft, Writing—review and editing; YJ, KHW, Conceptualization, Formal analysis, Validation, Investigation, Visualization, Writing—review and editing; KS, Conceptualization, Formal analysis, Supervision, Funding acquisition, Writing—original draft, Project administration, Writing—review and editing

### Author ORCIDs

Natalia Petrenko, http://orcid.org/0000-0002-2071-214X
Kevin Struhl, http://orcid.org/0000-0002-4181-7856

## Additional files

### Supplementary files

• Supplementary file 1. List of strains.

### Major datasets

The following dataset was generated:

| Author(s) | Year | Dataset title | Dataset URL | Database, license, and accessibility information |
|---|---|---|---|---|
| Petrenko N, Jin Y, Wong KH, Struhl K | 2017 | Mediator is essential for Pol II transcription, but is not a required component of the preinitiation complex | https://www.ncbi.nlm.nih.gov/geo/query/acc.cgi?acc=GSE93190 | Publicly available at NCBI Gene Expression Omnibus (accession no. GSE93190) |

The following previously published dataset was used:

| Author(s) | Year | Dataset title | Dataset URL | Database, license, and accessibility information |
|---|---|---|---|---|
| Paul E, zhu ZI, Landsman D, | 2015 | Saccharomyces cerevisiae S288c Genome sequencing | https://trace.ddbj.nig.ac.jp/DRASearch/study? | Publicly available via DNA Data Bank of |

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
