## [Decision Letter]

Thank you for resubmitting your work entitled "Evidence that Mediator is essential for Pol II transcription, but is not a required component of the preinitiation complex in vivo" for further consideration at *eLife*. Your revised article has been favorably evaluated by Jessica Tyler (Senior editor) and three reviewers, one of whom, Alan Hinnebusch, is a member of our Board of Reviewing Editors.

The manuscript has been substantially improved but there are a few remaining issues that we would like you to consider before acceptance, as outlined in the reviews of the revised paper below. Please pay particular attention to the criticism of the sentence in the Abstract concerning whether Mediator is required for PIC assembly raised by reviewer #3. Regarding reviewer #1's inquiry about the effect of TBP depletion on the shift in Pol II position (or lack thereof), as this was not raised in the previous round of reviews, it is not imperative that you address it in your final revision, but reviewer #2 agreed that to do so might help address his/her lingering skepticism about those results. It should be possible for the Reviewing Editor to examine the final revised paper without consulting with the reviewers.

*Reviewer #1:*

I am satisfied with the new experiments and revisions of text in this revised version of the paper. I am convinced by the authors' conclusions based on simultaneous depletion of head and tail subunits that Mediator is required for virtually all transcription in yeast cells. I am also satisfied with their alternative explanations for why depleting tail subunits simultaneously with depletion of an essential head subunit leads to a dramatic further reduction in transcription and PIC assembly compared to depletion of the head subunit alone: either reflecting an independent function of the tail module in PIC assembly or transcriptional activation that becomes critical when the head/middle domains are depleted, or results from loss of the ability of the tail domain to support recruitment of the small residual amounts of Head/Middle domains to the promoter. I also agree that the authors are justified in concluding that, in contrast to GTFs like TBP, the Mediator is not an essential, stoichiometric component of the PIC, such that PIC assembly and promoter escape can proceed in the presence of very low levels of Mediator at the core promoter in kin28-AA mutant cells, whereas only background levels of PIC assembly can proceed on depletion of TBP in the kin28-AA mutant. The latter conclusion rules out the possibility that the Mediator must provide a stoichiometric and persistent bridge between the enhancer (via the tail domain) and the PIC (via head and middle domains). Instead, the essential function of Mediator in PIC assembly could be exerted in a transitory or catalytic manner.

My only remaining question (in relation to Figure 3) is whether they observe any shift in the average position of Pol II relative to the TSS in the strain depleted of TBP? I believe they would predict that no shift should be observed, because the residual level of PICs that can be assembled at the depleted levels of TBP should behave normally with respect to Pol II escape from the promoter on CTD phosphorylation by Kin28. These results could be added to Figure 3 to provide an additional control for the specific effect of depleting Head/Middle subunits of Mediator.

*Reviewer #2:*

The revised version contains new experiments (notably the partial depletion of TBP) and better descriptions / discussion of the results, both contributing to make the work more intelligible. I particularly liked the new argumentation in the Discussion section. Hence, I consider that the current form of the manuscript merits publication. It will undoubtedly generate interesting discussions in the field. I am still not convinced about the Pol II shift, but this represents a minor part of the work overall.

*Reviewer #3:*

My main reservation with this paper, the claim that Mediator is not required for PIC formation, has been addressed-except in the Abstract, which states that: "PIC formation and Pol II transcription can occur when Mediator is not detected at core promoters". I believe that "occupancy" rather than "formation" is meant here. In fact, PIC formation and Pol II transcription can occur when Mediator is not detected at core promoters in wild type yeast, because of the rapid turnover of Mediator demonstrated by the Struhl and Robert labs earlier, as we have extensively discussed. So I don't understand this sentence as written. It will be confusing to many, maybe most, readers.

I think the result from Plaschka et al. (2015) should be mentioned. As Dr. Struhl recognizes, their result differs somewhat from what is reported in the manuscript under consideration. I would not say it is contradictory. Rather, the finding is that, after 18 minutes of incubation at 37˚C, srb4-ts mutant yeast exhibit an eight-fold decrease in nascent transcription, but this is partially offset by about a three-fold decrease in mRNA decay. This results in close to a 3-fold decrease in mRNA levels, consistent with the current work. I don't think this result contradicts, nor undermines the conclusions of the present work, as whether transcription rates are reduced to 33% or 12% of wild type levels, it still indicates function in the absence (or near absence) of Mediator. (I also had missed seeing this result until recently –that's why I didn't mention it in the first round of review – and I apologize for bringing it into the discussion a bit late, but I felt it was important to recognize.)

For the record, I believe the argument supporting the "missing piece of logic" is flawed. The argument is that if a GTF is depleted, transcription will decrease in proportion to the decrease in the GTF. This assumes that formation and function of the PIC is linearly dependent on GTF concentration, which seems unlikely. Depletion of individual Taf's, which I think Dr. Struhl would agree are GTFs, affects different promoters differently. Nonetheless, the new experiment is a good experiment, and the demonstration that GTF occupancy responds differently to depletion of Mediator as compared to depletion of other GTFs certainly indicates Mediator differs qualitatively (or very strongly quantitatively) in its properties with respect to PIC function and stability, which is really the point.

[Editors’ note: revisions had been previously requested before acceptance, as outlined below.]

Thank you for submitting your article "Evidence that Mediator is essential for Pol II transcription but is not a required component of the preinitiation complex in vivo" for consideration by *eLife*. Your article has been reviewed by three peer reviewers, one of whom, Alan Hinnebusch (Reviewer #1), is a member of our Board of Reviewing Editors. The evaluation has been overseen by Jessica Tyler as the Senior Editor.

The reviewers have discussed the reviews with one another and the Reviewing Editor has drafted this decision to help you prepare a revised submission.

All three reviewers felt that the manuscript has been improved and that the data in the paper are valuable in showing that Mediator is essential for virtually all transcription in vivo, and that appreciable levels of PIC assembly can occur in the presence of very low levels of intact mediator detectable at the promoter. However, all also felt that the paper ought to be re-written to avoid giving the impression that your data indicate that Mediator is dispensable for PIC assembly and merely stimulates the process. There was a unanimous opinion that the results in Figure 4 could be explained differently by proposing that PIC assembly on depletion of Med17 is still dependent on either the tail domain (known to be recruited on Med17 depletion) or by low levels of intact mediator (undetectable by ChIP) owing to incomplete anchor-away of Med17. The latter possibility of incomplete depletion of intact Mediator by depletion of Med17 alone is actually quite likely, in view of your findings that it has no effect on growth, and that a reduction of growth and strong impairment of PIC assembly and transcription requires the simultaneous depletion of tail subunits, which are known to be functional in recruitment. Thus, it seems imperative that you limit your conclusion to the statement that appreciable PIC assembly can occur in the absence of intact Mediator being detectable at promoters, and also clearly acknowledge the possibility that, while intact Mediator is not a required stoichiometric component of the PIC, it might still carry out an essential function in PIC formation that can be executed transiently or that requires only the tail domain.

Two of the reviewers also feel that your interpretation of the requirement for a triple depletion for a lethal reduction in transcription as indicating different functions for different modules is still too strong, and that it is equally if not more likely to simply reflect a more complete removal of Mediator from promoters owing to loss of recruitment activities of both head and tail modules simultaneously. While there is no requirement for you to adopt this view, it is asked that you please consider it once more in revising the paper.

*Reviewer #1:*

The authors have made considerable additions to the experimental results and extensive revisions of text that address all the important criticisms I had of the previous version of this paper. In particular, they documented the re-analysis of the med17-ts data from the Morse lab, and added additional experiments of their own on this mutant to confirm that transcription is attenuated, but not dramatically impaired, at most genes by this mutation, as they also found for the med17-AA mutant. They performed ChIP-seq analysis of Pol II in the triple mediator mutant to show that transcription is reduced genome-wide to the low levels seen on depletion of TBP, bolstering their conclusion that Mediator is on par with GTFs in terms of requirements for transcription. They included statistical analysis to show that the downstream shifts in Pol II profiles in certain med-AA mutants are statistically significant. They showed that depletion of Med17 in the kin28-AA does not completely eliminate TBP and TFIIB occupancies of the promoter, whereas depleting TBP in kin28-AA strain does so, which bolsters their conclusion that appreciable PIC assembly can occur in the absence of detectable intact mediator, and thus suggesting that intact Mediator is not an obligate stoichiometric component of the PIC. There are however a few issues that should be addressed:

1) In the experiment of Figure 4, they show that co-depleting Med17 and Kin28 does not reduce PIC assembly and transcription beyond the reduced levels conferred by depleting Kin28 alone, despite reductions of other Mediator subunits to background levels, and in Figure 4—figure supplement 3 they now show that depleting TBP in the kin28-AA strain evokes a much stronger reduction in PIC assembly and transcription, which together provide key evidence for their conclusion that Mediator is not a required component of the PIC, despite its established role in stimulating PIC assembly. Given that the tail domain is still recruited in the absence of Med17; and given their results from the triple mediator mutant implying that on depletion of Med17 alone either (i) mediator tail provides some function(s) in PIC assembly, or (ii) a functionally relevant albeit undetectable low level of intact mediator is still recruited, it seems important to state their conclusion about Mediator being dispensable for PIC assembly at bit more clearly. I think that they are justified in stating only that intact Mediator is not required at detectable levels at the UAS or promoter for appreciable PIC assembly; and to admit the possibility that the residual PIC assembly detected in Figure 4, and also inferred to exist by the downward shift of Pol II in Figure 3, could be dependent on the Mediator tail domain that is still recruited on Med17 depletion, or is dependent on a low level of intact Mediator that might still be recruited in the single med17-AA background that functions catalytically rather than as a stoichiometric PIC component. I feel that this more nuanced interpretation should appear in the Discussion section.

2) Figure 1—figure supplement 1: the Mediator subunit in light blue is missing in the key. It would also be helpful to remind the reader what subunits are in the tail, middle, or head (in the Results section).

3) Figure 4—figure supplement 1: the color key is missing.

4) Subsection “Pol II transcription can occur from preinitiation complexes lacking Mediator”: cite specific figure panels that allow comparisons of Pol II and/or TFIIB occupancies at the same genes in kin28-AA tbp1-AA double mutant vs kin28-AA med17-AA double mutant that must be inspected to justify the conclusion that depleting TBP has a stronger effect on PIC assembly than does depleting Med17 in the kin28-AA background.

*Reviewer #2:*

The manuscript by Petrenko et al. describes the effect of Mediator subunit depletions by anchor-away on Mediator, RNAPII and GTFs occupancy. Their conclusions can be summarized as follow: The Mediator Core (defined as Head and Middle) and Tail make additive contributions to the role of Mediator in transcription. None of these two functions is essential on its own, but their additive disruption leads to the inability to transcribe. Mediator is therefore described as essential but through two non-essential functions. This contrasts with the current view of an essential Core and a dispensable Tail. In addition, the work provides additional insights such as i) a role for Mediator in the inhibition of promoter escape and ii) in vivo support to the stimulation of TFIIH kinase by Mediator.

The experiments are well executed, and care is taken trying to control for everything but the study is suffering from the inherent limitation of the anchor-away technique: incomplete depletion cannot be ruled out. In addition, a weakness resides in the fact that key conclusions are based on negative ChIP data (i.e. failure to detect Mediator at core promoters is interpreted as a complete absence of the complex). Yet, failure to detect a protein by ChIP could always be interpreted as presence but under the detection limit. This is particularly relevant for Mediator given the inherent transitory nature of the interactions it makes with UAS and (especially) promoter regions. Previous work by the authors represents a good demonstration of that concept: in WT cells, Mediator is not detectable at promoters, yet it is now though to be there, but too transiently to be detected by ChIP. For the conclusions stated by the authors to stand, one would have to rule out the following (and perhaps other) scenario:

Core Mediator (Head and Middle) is essential for transcription and is an essential component of the PIC (the alternative model to the one proposed by the authors) but the observations presented in the current manuscript are explained by the following: Depletion of Mediator subunits is incomplete (as acknowledged by the authors) and minutes amount of intact core Mediator and/or remaining core Mediator sub-complexes are able to nucleate enough PICs to sustain reasonable amount of transcription; aided by the disconnected Tail bound at enhancers. When, in addition, the Tail is removed, the remaining parts no longer manage to sustain PIC assembly. The arguments in the manuscript against this scenario reside in the fact that Mediator occupancy at core promoters (as measured in kin28-AA conditions and shown in Figure 4) is decreased to background levels upon med17-AA. While this illustrates the important contribution of Med17 on Mediator integrity and function, it does not rule out the possibility that Mediator or Mediator parts make transient contacts with the PIC.

The model proposed by the authors is also inconsistent with the observation that most Head and many Middle subunits are essential. The fact that anchoring-away these essential subunits does not prevent growth is the proof that depletion is incomplete. Otherwise, how could these cells grow? The proposition that the essential role of these proteins resides in the cytoplasm is not supported by evidence. The authors argue that because anchoring away GTFs leads to loss of growth, the ability to grow after depletion of essential Mediator subunits is unlikely to be due to incomplete depletion. It appears likely to this reviewer that completely removing Mediator would be more difficult than depleting GTFs. GTFs are, for most of them, single subunit or small complexes. Mediator, on the other hand, is made of 25 different polypeptides. Also, it is not very difficult to imagine that higher amounts of GTFs than Mediator are required for transcription. GTFs are stable components of the PIC and many may even stay in place after escape as a scaffold. This is very different from Mediator, which very transiently associates with the PIC, as the authors themselves showed in a previous publication.

This reviewer is also not convinced about the shift in Pol II distribution shown in Figure 3. This effect is very small and it is not clear that it can be interpreted as a faster escape. It looks like the curves have been re-scale so that they all reach the same level at the 0.5kb position (please confirm). This alone may introduce distortions in traces like the one depicted here. Also, given that Mediator stimulates TFIIH-mediated phosphorylation of the CTD, the interpretation made by the authors here is very counter-intuitive.

In sum, the model proposed by the authors may very well be right but alternative explanations are just as likely to be right. The data presented in this manuscript has value but the interpretation that is made of it goes too far.

*Reviewer #3:*

This revised manuscript addresses most of the major criticisms of my previous review. In particular:

1) The new experiment, in which Pol II genome-wide occupancy is compared in TBP-AA and triple depletion strain (Figure 6), provides strong evidence for an absolute, or near absolute, requirement for Mediator for transcription in vivo.

2) Results bearing on the requirement of Mediator as a PIC components are strengthened by additional data, particularly measurements of TBP and TFIIB following Med17 depletion. However, although I agree that the results presented provide strong evidence that Mediator is not a required component of the PIC, I do not agree that the authors have shown that Mediator is not required for PIC formation. More on this below.

3) The additional data on the downstream of Pol II (Figure 3) is helpful, as is the statistical analysis.

4) The revised discussion regarding "independent contributions" of Mediator modules has somewhat tempered what was a radical proposal based on slim data. I still believe the discussion on this point to be a bit slanted (i.e. I would favor the idea that tail module-dependent recruitment of the middle and possibly head module, at low levels, may be responsible for the residual activity seen in the med17-AA strain), but that's an opinion.

5) The finding that depletion of various Mediator subunits from the middle and head modules show stronger effects on Pol II occupancy at SAGA-dependent than at TFIID-dependent genes is unexpected and very interesting. It should be mentioned in the Abstract.

I do still have one important point that I would like to see the authors address, along with several smaller issues. The major point is, as mentioned above, that the evidence does not support the contention that Mediator is not required for PIC formation. The Abstract states the findings accurately: "Mediator is essential for Pol II transcription and stimulates PIC formation, but it is not a required component of the PIC in vivo." This seems reasonable; it could be argued that we already knew this, since Mediator appears only to be present transiently at the PIC in wild type yeast, whereas GTFs and Pol II can be detected readily by ChIP. But the Discussion includes a section with the heading "Mediator stimulates, but is not required for PIC formation in vivo". The arguments for this point are: 1) the low Mediator:GTF ratio seen upon depletion of Med17 and Kin28, which they claim "provides very strong evidence of Mediator-independent PIC formation and function" – function, yes; formation, not so much; 2) the downstream shift of Pol II occupancy (Figure 3) – I don't understand how this is pertinent to the question of PIC formation, which might occur in the presence of low levels of Mediator (hence the low Mediator:GTF ratio); and 3) the finding that anchor-away of Mediator subunits allows growth, whereas depletion of GTF components does not.

I think several points argue against this conclusion. First, the dichotomy between Mediator subunit depletion and GTF depletion is strong evidence that depletion is incomplete, as are the results shown in Figure 6, which show that depletion of TBP or Rpb1 does not completely eliminate Pol II occupancy. Evidently yeast can tolerate low levels of Mediator and remain viable, whereas low levels of TBP or Pol II present after depletion, although not completely eliminating transcription, do not support life. The authors argue that observations that Pol II and GTF occupancy is reduced less strongly than Mediator occupancy (Figure 4) implies that Mediator is not needed for PIC formation. Mediator depletion is clearly not complete, and further depletion (as in Figure 6, using the triple depletion strain) does reduce Pol II occupancy to the same level, approximately, as seen in Tbp-AA or Rpb1-AA strains. Why does this not imply that Mediator is needed for PIC formation, and that the occupancy seen in kin28-AA med17-AA yeast is due to residual Mediator activity? Perhaps the argument is based on an assumption that Mediator must be retained at stoichiometric levels at the PIC if it is recruited along with the PIC in the absence of Kin28 function. If that assumption were true, then it would indeed follow that the evidence shows that Mediator is not needed for PIC formation. But I don't know what evidence there is for such an assumption, and the assumption should at least be stated.

In summary, I do think the findings are of broad interest and well documented. They shed considerable light on old data regarding transcription occurring independently of (actually in med17-ts yeast) Mediator, and provide new insight into the role of Mediator in PIC formation (needed only transiently, perhaps not at all) and transcription. A more rigorous interpretation of results would, in my opinion, enhance impact.

[Editors’ note: a previous version of this study was rejected after peer review, but the authors submitted for reconsideration. The first decision letter after peer review is shown below.]

Thank you for choosing to send your work entitled "Evidence that Mediator is essential for Pol II transcription but is not a required component of the preinitiation complex in vivo" for consideration at *eLife*. Your article has been considered by a Senior Editor and three reviewers, one of whom, Alan Hinnebusch, is a member of the Board of Reviewing Editors.

Our decision has been reached after consultation between the reviewers. Based on these discussions and the individual reviews below, we regret to inform you that your work will not be considered further for publication in *eLife*.

The first main conclusion of the paper, that Mediator is essential for Pol II transcription in yeast, was generally accepted by the reviewers but with a number of important qualifications. To establish the novelty of the findings, it seems necessary for the authors to document their re-analysis of data from the Morse lab to justify their claim that Pol II occupancies were reduced only modestly by the med17-ts mutant in that study. The second issue involves a lack of thorough documentation of the reductions in Mediator and Pol II occupancies in the single versus triple Mediator subunit deletions at a large group of genes. In Figure 1, PMA1 and CCW12 were the only two genes for which both Mediator and Pol II occupancies were reported, and it's important to conduct Mediator ChIP analysis at HSP82 and CUP1, where transcription was reduced considerably less than at PMA1. It is also critical to extend the analysis of the triple mutant in Figure 6 to include genome-wide Pol II ChIP data, and compare it to anchor-away of TBP in order to demonstrate the occurrence of similar strong reductions in transcription genome-wide on depletion of either three Head/Tail Mediator subunits or the GTF TBP. This will address whether or not there is a significant fraction of genes behaving like HSP82 and CUP1 in Figure 6, wherein depletion of Mediator produces a significantly smaller reduction in Pol II occupancy versus depletion of TBP (Figure 6).

To account for the greater effect on transcription for the triple versus single Mediator mutants, the authors propose independent functions for the Head/Middle and Tail domains. However, the authors should discuss alternative explanations for these results, including the possibility that, realizing that anchor-away of Med17 in the single mutant is almost certainly incomplete (no growth defect at 30C), the removal of Med17 from promoters is simply rendered more complete in the triple mutant owing to the loss of the tail subunits and their known function in Mediator recruitment; alternatively, the triple mutation might reduce occupancy of the Middle module to a much greater extent than do any of the single subunit mutations, and it is loss of Middle domain functions that is involved in the triple mutant. A related issue is that, in attempting to explain how seemingly complete depletion of single Mediator subunits produces no growth phenotypes, the authors unjustifiably dismiss incomplete depletion as the most likely explanation for the result, and do not consider that a very low or transient association of Mediator with the promoter below the detection limit of ChIP, is sufficient for a minimum amount of transcription of essential genes required to sustain growth.

The evidence supporting the second major conclusion, that while being essential for transcription, Mediator is not required for PIC assembly, was criticized to a greater extent. One of the lines of evidence, of the shift downstream in Pol II occupancies, has been criticized on several grounds: a statistically significant effect is unlikely to have been observed for all three mutants analyzed in Figure 3; and the effect may be an artifact of scaling of the data in different mutants. The other line of evidence in Figure 4, showing that in kin28-aa cells where Mediator should accumulate at the promoter, depletion of Med17 evokes strong reductions in Mediator occupancy of promoters but minimal reductions in transcription. There are several criticisms of this experiment. One is that kin28-AA alone reduces transcription considerably and it's unclear whether they could measure a further reduction on co-depletion of Med17 at most of the genes analysed. It seems necessary to show that co-depletion of TBP in the kin28-AA mutant would evoke a much greater decline in transcription than was observed for co-depletion of Med17 to bolster their interpretation, preferably with genome-wide ChIP-seq analysis of Pol II. Other comments raise the point that because Mediator association with promoters is known to be dynamic, the Mediator:Pol II ratio at the promoter may not be the most incisive measure of Mediator function. Two reviewers insist that it is necessary to provide independent support for their key conclusion by showing that TBP is still recruited normally to promoters in the triple Mediator mutant. Even if this important line of evidence was obtained, the authors would still be urged to interpret the data cautiously and, given that no depletion is equivalent to a null allele, admit that they cannot dismiss the possibility of a transient, perhaps catalytic, role of Mediator that can be accomplished without stable, detectable Mediator occupancy at promoters.

Finally, the authors may need to reinterpret their data on SAGA- versus TFIID-dependent genes, and Mediator occupancies at RP and glycolytic genes, in light of other relevant publications.

The amount of additional work being requested to address the reviewers' criticisms is considerable, and it is unclear whether the results of these new experiments would support the authors' main conclusions. Thus, it is impossible to judge the work as potentially acceptable with suitable revision. However, this same group of referees would be willing to consider an extensively revised manuscript containing significant additional experimentation and analyses that would address all of their serious concerns.

*Reviewer #1:*

This paper employs the anchor away technique to deplete subunits of Mediator in yeast cells and examine the consequences on transcription. Previous work using a med17-ts allele had shown that bulk poly(A) mRNA is dramatically reduced at the restrictive temperature, leading to the conclusion that Mediator is strongly required for transcription of the majority of genes, although some exceptions of stress-induced genes have been observed. The authors have re-analyzed more recent genome-wide data on the med17-ts mutant from the Morse lab and claim that Pol II occupancies are reduced globally but by only a factor of 2-3; raising the question about whether Mediator is truly as important as GTFs for transcription in vivo. They show that depleting any single Mediator subunit, including Med17, has less effect on transcription genome-wide (as assessed by Pol II occupancies in coding sequences (CDS)) compared to depleting TBP, despite reductions in Mediator occupancy to background levels at the few (three) genes examined by ChIP. This is true for both constitutive and induced genes; although the PMA1 gene does show a dramatic loss of transcription in the med17-AA strain, as shown previously for this gene in a med17-ts mutant. They find that SAGA-dependent genes exhibit greater reductions in transcription versus TFIID-dependent genes on depletion of Mediator head, middle, or tail domain subunits, which is consistent with previous results from the Morse lab on elimination of tail subunits. They further show a ~50bp (by my calculation) shift in average Pol density downstream from the TSS on depletion of Med14 or -17, but not on depletion of tail subunits, which is consistent with previous conclusions that Mediator-Pol II interaction at the promoter inhibits promoter escape prior to CTD phosphorylation by Kin28 of TFIIH, and that head subunits can be recruited to promoters at least to some extent in tail mutants. This downstream shift is also presented as evidence that PIC assembly is occurring at promoters on depletion of these head or middle subunits and that the assembled PIC escapes at a higher frequency owing to loss of the Mediator Head/Middle connection with Pol II at the promoter. This interpretation seems justified if the downstream shift is found to be statistically significant. In an effort to bolster the conclusion that PIC assembly can occur without Mediator at the promoter, they deplete Mediator subunits in a kin28-AA strain, exploiting the previous findings that depleting Kin28 prolongs Mediator occupancy at the promoter by preventing Pol II escape and attendant dissociation of Mediator from Pol II. They again find a strong reduction in Mediator occupancy at the five different promoters tested on depletion of Med17, but see no further reduction in transcription from that seen in the kin28-AA single mutant. They infer from this result that Mediator is not an essential component of the PIC; however, the interpretation of these results is not straightforward, as discussed further below. A different result was obtained when they depleted Med17 in a double mutant lacking the tail subunits Med3 and Med15, which reduced transcription further than seen from deletion of the tail subunits alone, at six different genes. In addition, they found that the depletion of individual Mediator subunits had no effect on cell growth, unless carried out at 37C, but that depletion of Med17 in the strain lacking the two tail subunits strongly reduced growth. They interpret these results by proposing that the tail and head/middle modules of Mediator have distinct functions, and that one or the other must be retained for appreciable transcription. This leads them to conclude that Mediator is indeed essential for transcription, even though they consider it dispensable for PIC assembly. Finally, they present evidence that depletion of Mediator subunits reduces Ser5 CTD phosphorylation, providing in vivo evidence for this function of Mediator that was demonstrated long ago in vitro.

General critique:

The conclusion that Mediator is essential for transcription in vivo, based on the results of simultaneous deletion/elimination of three Mediator subunits seems well demonstrated by the results in Figure 6. These data represent a more definitive finding than the original results of Young et al. that were based on mRNA measurements alone and did not rule out secondary effects on mRNA stability; and which are apparently in conflict with more modest effects on transcription observed by the authors in re-analyzing published data from the Morse lab. It seems necessary for the authors to document their re-analysis of the Morse data to justify this claim. Moreover, their interpretation of the additive effect of combining head and tail deletions as evidence for distinct functions of these modules is not compelling as it overlooks the possibility that the removal of Med17 from promoters is simply more complete in the triple mutant owing to the loss of the tail subunits and their known function in Mediator recruitment by activator proteins. The authors acknowledge that anchor-away of Med17 is unlikely to be complete, and it has been demonstrated previously that tail subunit deletions reduce the occupancy of head/middle subunits, at least at certain promoters (see specific comments below for details). As such, it is entirely possible that there is a functionally significant level of Med17 recruitment, albeit below the detection limit of their ChIP assays, that is retained in the med17-AA single mutant but reduced further or even eliminated in the triple mutant owing to loss of the established role of tail subunits in Mediator recruitment to enhancers. It seems that this interpretation should be included in the Discussion as an alternative to their suggestion that the tail has a distinct function beyond Mediator recruitment, for which they cite no other evidence.

The second important conclusion, that PIC assembly can proceed without Mediator, is a plausible interpretation of the results in Figure 3, which imply enhanced Pol II release from PICs assembled at low levels of Med17. However, it is important to establish the statistical significance of this shift. The supporting results in Figure 4 are more complicated, however, because the kin28-AA mutation reduces transcription dramatically on its own, making it difficult to determine whether or not co-depletion of Med17 could evoke a further decline that could be measured at four of the five genes they analyzed; and it seems necessary to establish that co-depletion of a GTF (e.g. TBP) would reduce transcription further in the kin28-AA cells, in contrast to depleting Med17. In addition, it's difficult to understand why co-depletion of Med17 would not reduce PIC assembly whatsoever even if Mediator is not essential for this process unless Mediator is completely uninvolved in recruiting Pol II, which seems unlikely. It is worth noting that it was shown previously that TBP recruitment to Gcn4 activated promoters is impaired in single mutants lacking either a head or tail subunit (Qiu, H., et al. (2004). Mol Cell Biol 24(10): 4104-4117.); and there may be similar such measurements published for other activated genes. Hence, to bolster their conclusion that Mediator is not a required component of the PIC, it is important that they attempt to demonstrate this point more directly by analyzing TBP or TFIIB promoter occupancies on Med17 depletion, in both WT cells and in the double mutant lacking the tail subunits Med3 and Med15. If their thesis is correct, appreciable TBP/TFIIB recruitment should be maintained in the absence of these Mediator subunits.

Finally, it's unclear whether they believe that Mediator has no role in PIC formation or that it is simply not required as a stable, stoichiometric constituent of the PIC. If it is not required for PIC formation, then what is its essential role in transcription? Is it enhancing Kin28 function to allow promoter escape? Is that essential? The paper should discuss what the essential role of Mediator could be if it is not to stimulate PIC assembly.

Specific comments:

Introduction section: the re-analysis of the Pol II ChIP-chip data published by Paul et al. should be presented as a detailed supplementary figure to substantiate this claim.

Results section: the reduction in transcription at PMA1 on depletion of Med17 is not modest, so this claim should be qualified to describe the average behavior.

Results section and Figure 1: PMA1 is the only gene for which both Mediator and Pol II occupancies were measured, and both occupancies are dramatically reduced by Med17 depletion. It seems important to examine Med17 occupancies of all of the genes whose Pol II levels were examined in this figure, rather than simply assuming that every gene exhibits the same strong reduction shown for CCW12 and SED1.

Figure 1 and Figure 2: it seems important to acknowledge that the AA-tags introduced into Mediator subunits reduce their functions appreciably in the absence of rapamycin.

Figure 2: the color-coding should be labeled as log_2_ [(Pol II(-rap)/Pol II(+rap)] or something similar.

Figure 3: it is important to establish the statistical significance of the shift in Pol II downstream.

Subsection “Pol II transcription can occur from preinitiation complexes lacking Mediator” and Figure 4: the interpretation of these data is complicated by the fact that transcription is reduced extensively at four of the genes by depletion of Kin28 alone, making it difficult to determine whether a further reduction ensues with Med17 depletion. Would they be able to see a further reduction if they depleted a factor essential for PIC assembly, e.g. TBP, in the kin28-AA mutant? This control would seem to be essential to support their interpretation. In addition, even if Mediator is not essential for PIC assembly, which seems reasonable to conclude for CUP1, wouldn't they expect that Med17 depletion would reduce PIC assembly at this gene by decreasing Pol II recruitment? One way of interpreting these data is to conclude that Med17's role in stimulating transcription at these genes is completely dependent on Kin28, which in turn could mean that Med17 acts primarily by enhancing Kin28 function in promoter escape (kin28-AA is epistatic to med17-AA)? It also seems difficult to eliminate a role for Mediator in PIC assembly where it would not have to function as a stable, stoichiometric component of the PIC, i.e. more like an enzyme. In fact, significant Med17 occupancy is detected at HSP82 and CUP1 following depletion of Med17 in the kin28-AA strain-perhaps this low level is sufficient.

Subsection “Pol II transcription is virtually eliminated when Mediator head, middle, and tail modules are simultaneously inactivated” second paragraph: shouldn't this read: "…in the double mutant strain…(Figure 6)"?

“Thus, depletion of all three Mediator modules has a stronger transcriptional effect than conditions where the tail module is present at enhancers (Med17 depletion) or the head and middle modules are present at core promoters (deletion of tail subunits)”: this statement should be revised as it seems to imply that deletion of the tail subunits does not reduce Med17 occupancy, whereas this is very unlikely, based on previous studies showing a strong reduction in Head subunits when the tail is deleted (e.g. the reference Zhang et al. (2004)). Also tail subunit occupancies could be reduced, even if not eliminated, by depletion of Med17.

“As deletion of some Mediator subunits […] grow at 37C (Figure 7)”: there are several parts of this sentence that refer to published findings that are not cited.

Subsection “Growth of Mediator-depletion strains” paragraphs two and three: it seems possible that depletion of Mediator subunits is less deleterious than depleting GTFs because Mediator can perform one or more functions catalytically at much lower cellular levels without being a stable, stoichiometric constituent of the PIC, e.g. stimulating Kin28 kinase activity.

*Reviewer #2:*

In this manuscript, Jin, Struhl, and colleagues report "Evidence that Mediator is essential for Pol II transcription but not a required component of the preinitiation complex in vivo". Although the results presented are interesting, I don't believe they are conclusive with regard to the strong statements made in the title and Abstract.

Much evidence exists in the literature for the importance of Mediator in mRNA transcription. Thus, the distinction between Mediator being important and essential is critical for the impact of this manuscript. The evidence that Mediator is essential for Pol II transcription is based on the "triple" strain, med3∆ med15∆ med17-AA, in which Med17 is depleted from the nucleus using the anchor away technology. Figure 6 shows that there is indeed a strong effect on Pol II occupancy in this strain. However, the effect is not complete, as the authors acknowledge. In fact Pol II is still enriched at CUP1 by about 15 fold, or about four fold less than in WT yeast, and close to ten-fold at HSP82, about a five-fold reduction from wild type levels. The authors interpret this as being more likely to be caused by incomplete Mediator depletion than by activation without Mediator by comparing the reduction of Pol II occupancy after depleting Rbp1 or TBP using anchor away. Although they state that the effects are "roughly comparable to (although perhaps slightly lower than) that occurring in the triple deletion strain", it seems clear that the effect really is less. The graphs for the triple deletion strain ought to be presented side by side with the rpb1-aa and tbp-aa strains; it's not clear that "perhaps" applies, and therefore the argument that Mediator is truly essential-that transcription absolutely depends on its presence-is weakened. In addition, given the importance of this result, Pol II occupancy ought to be measured genome-wide as it was for individual anchor-away experiments in earlier figures.

The second principal conclusion of the paper is that Mediator is not a required component of the PIC in vivo. This is based on experiments in which depletion of Med17 by anchor away, or inactivation in the classical med17-ts mutant, results in only partial loss of Pol II association with ORF regions while Mediator occupancy at promoters (seen by also anchoring away Kin28) is greatly reduced. Reduction of Mediator occupancy is only shown for three FRB-tagged Mediator subunits at the CCW12 enhancer and for med17-aa at three promoters. Depletion should be examined genome-wide for at least some of the anchor-away strains, probably best while also depleting Kin28. In addition, the authors emphasize that the low Mediator:Pol II occupancy ratio at the core promoter seen when both Med17 and Kin28 are depleted "provides very strong evidence of Mediator-independent PIC formation and function". But measurement of this same ratio in KIN28+ yeast would also yield a very low Mediator:Pol II ratio; is it not therefore possible that dynamics still play a role in this measurement and that it does not provide a completely accurate picture of PIC composition in vivo? In addition, once Pol II escapes the promoter, it will continue to contribute to ChIP occupancy measurements but will no longer be part of a PIC as normally understood. It would be more convincing to also measure occupancy of other PIC components such as TBP or TFIIB in this experiment and compare them to Mediator occupancy; but even if this were done, questions of dynamics with regard to Mediator would persist.

Another piece of evidence cited for Mediator not being required for PIC formation in vivo is the downstream shift of Pol II seen upon depletion of Mediator head or middle subunits. However, this shift (Figure 3) is observed most strongly for med7-aa yeast, somewhat less strongly for med14-aa, and not at all for med17-aa. Thus, the evidence for this downstream shift of Pol II is ambiguous. Also, a downstream shift in Pol II was reported upon Kin28 inactivation by Rodriguez-Molina et al. (2016) Mol. Cell 63:433, and interpreted as representing a yeast-specific elongation checkpoint. The authors should discuss this result in light of their own findings.

A third major conclusion is that Mediator modules make independent contributions to the overall transcriptional function of Mediator (Discussion section). If I understand the argument correctly, the authors suggest that in med17-aa yeast, association of the tail module of Mediator with enhancers is sufficient to activate many genes in the absence of middle/head module function. Since there is no evidence for such independent function (albeit independent recruitment has been demonstrated) of the tail module, it seems more economical to postulate that decreased function of the middle/head module, or continued function of the middle module (which does make contact with PIC components), accounts for remaining activity. The authors also appear to argue that the "apparent absence of Mediator at enhancers that drive expression of ribosomal protein and glycolytic genes" could be due to the middle/head modules functioning independently of the tail module at these genes. But low Mediator signal at these signals actually appears to be caused by the same Kin28-Pol II CTD mediated dynamics that the authors and the Robert group have reported, and that is used here to advantage. For example, see Jeronimo and Robert (2014) Figure 3D and Supplementary Figure 2B.

*Reviewer #3:*

The manuscript by Jin et al. provides two fundamental conclusions about Mediator in yeast cells: 1) Mediator is essential for transcription in vivo and 2) Mediator is not a required component of the PIC in vivo.

The first conclusion was somewhat expected (at least for most people) since a frequently cited study from the Young lab has shown that a ts mutant for Srb4/Med17 leads to massive decrease in steady state mRNA levels. As well articulated by the authors, this old data was not directly addressing transcription so a formal demonstration of the essentiality of Mediator for transcription in vivo was lacking. The authors addressed this question by combining a double deletion of Tail module subunits with the nuclear depletion of the head subunit Med17. This triple mutant was used in Pol II ChIP assays to show decrease in Pol II occupancy comparable to those observed when depleting TBP or Pol II itself. This indeed provides compelling evidence that Mediator is essential for transcription in vivo.

The second conclusion, however, is more surprising and also not as decisively supported by the data. In sum, they showed that depletion of individual subunits (Tail, Head or scaffold) -unlike the triple mutant- leads to only partial (2-3 fold) reduction in Pol II occupancy over genes. Because Head and Middle subunits can interact with the PIC in the absence of a Tail, and because the Tail can still be recruited to enhancers in the absence of Head and Middle, they interpret this data to say that Tail and Head/Middle have independent contributions to transcription, none of them being essential on its own. They then looked at Mediator occupancy at promoters in conditions when it is detectable (Kin28-depletion) to show that it decreases to a much larger extent than Pol II upon Med17 depletion. Such a low Mediator:Pol II ratio at promoters in Med17-depleted cells is interpreted as a strong indication that a PIC can form in the absence Mediator in vivo. Although they acknowledge the fact that Med17 depletion is likely to be incomplete, the authors argue that their conclusion can be made independently of a complete depletion. Although I suspect that the authors' conclusion is correct, I think that alternative interpretations cannot be completely excluded. Mediator occupancy at core promoters is very transient. In addition, sub-Mediator complexes have been shown to exist in cells. Could it be that upon depletion of Med17, assembly of Mediator within the PIC still occurs but in a less stable (more transient) manner. This would lead to a decreased Mediator-Pol II ChIP ratio but would not rule out the possibility that this transient interaction is nevertheless necessary for transcription. In essence, Med17 depletion would simply exacerbate a phenomenon already present in WT cells: A transient but necessary interaction of Mediator with the PIC. Again, this is perhaps not the most likely explanation, but I do not think one can rule it out completely. The authors provide evidence to dispute it or if not, acknowledge this possibility and modify the title of their manuscript accordingly.

Other comments:

It is stated in the Introduction and later in the Discussion that Mediator poorly (if at all) associates with the enhancer of RP and glycolytic genes. While this is indeed what is observed by ChIP, a recent paper used ChEC-seq to show that Mediator can be found at virtually all enhancers in yeast, including those upstream of RP and glycolytic genes. This suggests that Mediator detection by ChIP is likely to be dependent on its dwell time on enhancer DNA, similarly to what was shown on promoter DNA. This paper should be cited and the argumentation should be modified accordingly.

The data shown in Figure 3 is interesting and in line with a recent study from the Malik lab (PMID: 27916598). This should be cited and perhaps discussed. Although interesting, this analysis is rather thin and not very convincing. Indeed, the shift downstream is very small and required scaling the different datasets. While this is a reasonable thing to do to the data, I am afraid that the shift may be an artifact of the scaling. Can the author further this analysis? Perhaps they could find a group of gene where the effect is more readily visible and show specific examples. Also, can they provide an analysis ruling out the possibility that the shift correlates with the amount of scaling applies to each dataset?

As mentioned by the authors, the fact that depletion of essential Mediator subunits does not abrogate growth is puzzling. The paragraph on this topic (in the subsection “Growth of Mediator-depletion strains”) is not very compelling. The authors claim that incomplete depletion is unlikely to be the explanation, yet they do not provide any alternative (except for a "non-chromosomal" function, which would be extremely surprising and for which there is absolutely no evidence). Their argument to dismiss incomplete depletion is the fact that depletion of Kin28 is lethal despite a comparable effect on Pol II. This is not a valid argument since it is well established that kin28 lethality is due to defect in post-transcriptional events such as mRNA capping. With the lack of a better explanation, the authors should acknowledge incomplete depletion as the most likely explanation. This also implies, that very little amount of Mediator is necessary for growth, and hence for transcription, which is in line with their ChIP data.

Regarding the Mediator-dependent at SAGA versus TFIID genes, it has recently been argued that this may be due to technical limitations (PMID: 27773677). Can the authors comment on that?

It would be important to base the conclusions on PIC on a more direct measure of PIC assembly.

---

## [Author Response]

*Reviewer #1:*

*[…] My only remaining question (in relation to Figure 3) is whether they observe any shift in the average position of Pol II relative to the TSS in the strain depleted of TBP? I believe they would predict that no shift should be observed, because the residual level of PICs that can be assembled at the depleted levels of TBP should behave normally with respect to Pol II escape from the promoter on CTD phosphorylation by Kin28. These results could be added to Figure 3 to provide an additional control for the specific effect of depleting Head/Middle subunits of Mediator.*

We had already done the analysis for Pol II profile under TBP-depletion conditions, and the result is as expected, namely no Pol II shift. We originally chose not to include it because we felt that readers might get confused seeing a Pol II profile under conditions where there is theoretically no transcription (of course there is some due to incomplete depletion). It would be particularly confusing when we normalized everything to 100% since it would look like there were wt-levels of transcription. So, in this latest version, we have included the data as Figure 3—figure supplement 1 so that it doesn’t get confused with the Mediator data.

*Reviewer #3:*

*My main reservation with this paper, the claim that Mediator is not required for PIC formation, has been addressed-except in the Abstract, which states that: "PIC formation and Pol II transcription can occur when Mediator is not detected at core promoters". I believe that "occupancy" rather than "formation" is meant here. In fact, PIC formation and Pol II transcription can occur when Mediator is not detected at core promoters in wild type yeast, because of the rapid turnover of Mediator demonstrated by the Struhl and Robert labs earlier, as we have extensively discussed. So I don't understand this sentence as written. It will be confusing to many, maybe most, readers.*

We changed the sentence, although differently than suggested. The suggestion of “PIC occupancy” makes no sense because a PIC can’t occupy anything or be occupied by anything. I can see how readers could be confused by PIC formation (even though our original statement was accurate), so we changed it to a “functional PIC and transcription”, which makes the point better.

Reviewer 3 is still confused about the rapid turnover of Mediator. In our paper on this, we specifically defined the PIC as a Mediator-containing entity based on the belief at the time that Mediator was a GTF. As such, the Mediator-lacking entity at the core promoter was termed a “post-escape complex” that was transcriptionally inactive. At the time, we never considered or discussed the idea of a Mediator-lacking PIC, but that is the main point of the present paper.

*I think the result from Plaschka et al. (2015) should be mentioned. As Dr. Struhl recognizes, their result differs somewhat from what is reported in the manuscript under consideration. I would not say it is contradictory. Rather, the finding is that, after 18 minutes of incubation at 37˚C, srb4-ts mutant yeast exhibits an eight-fold decrease in nascent transcription, but this is partially offset by about a three-fold decrease in mRNA decay. This results in close to a 3-fold decrease in mRNA levels, consistent with the current work. I don't think this result contradicts, nor undermines the conclusions of the present work, as whether transcription rates are reduced to 33% or 12% of wild type levels, it still indicates function in the absence (or near absence) of Mediator. (I also had missed seeing this result until recently –that's why I didn't mention it in the first round of review – and I apologize for bringing it into the discussion a bit late, but I felt it was important to recognize.)*

We now cite Plaschka et al., 2015 as requested. I still think this previous paper is somewhat contradictory Paul et al. and our work. Reviewer 3 suggests that the increased halflife of mRNAs in the med17-ts mutant compensates for the apparently greater loss of nascent transcription. However, we are measuring Pol II occupancy, so increased mRNA stability should not be relevant. One would think that Pol II occupancy and nascent transcription should give similar results, although maybe this isn’t strictly true and in any event it is difficult to know where the errors are in Plaschka et al., and the apparent difference is only about 2-fold.

*For the record, I believe the argument supporting the "missing piece of logic" is flawed. The argument is that if a GTF is depleted, transcription will decrease in proportion to the decrease in the GTF. This assumes that formation and function of the PIC is linearly dependent on GTF concentration, which seems unlikely. Depletion of individual Taf's, which I think Dr. Struhl would agree are GTFs, affects different promoters differently. Nonetheless, the new experiment is a good experiment, and the demonstration that GTF occupancy responds differently to depletion of Mediator as compared to depletion of other GTFs certainly indicates Mediator differs qualitatively (or very strongly quantitatively) in its properties with respect to PIC function and stability, which is really the point.*

I don’t think the “missing logic” experiment is flawed. Reviewer 3 is correct that the levels of PIC assembly and transcription might not be linear with TBP levels, but this is not what we are saying or implying. The point is that PIC levels (GTF occupancy) goes with transcription at all promoters, so incomplete depletion doesn’t change the occupancy ratios of GTFs. If one reduces overall TBP levels by 2-fold, this doesn’t mean that PIC levels change 2-fold. Instead, we are saying that if PIC levels drop 2-fold, Pol II transcription drops 2-fold. Reviewer 3 is correct that promoters will behave differently at intermediate TBP levels in terms of PIC and transcription levels, but the occupancy ratios of GTFs and transcriptional output will still go together.

[Editors’ note: the authors’ response to the previous round of review follows.]

*[…] Reviewer #1:*

*The authors have made considerable additions to the experimental results and extensive revisions of text that address all the important criticisms I had of the previous version of this paper. In particular, they documented the re-analysis of the med17-ts data from the Morse lab, and added additional experiments of their own on this mutant to confirm that transcription is attenuated, but not dramatically impaired, at most genes by this mutation, as they also found for the med17-AA mutant. They performed ChIP-seq analysis of Pol II in the triple mediator mutant to show that transcription is reduced genome-wide to the low levels seen on depletion of TBP, bolstering their conclusion that Mediator is on par with GTFs in terms of requirements for transcription. They included statistical analysis to show that the downstream shifts in Pol II profiles in certain med-AA mutants are statistically significant. They showed that depletion of Med17 in the kin28-AA does not completely eliminate TBP and TFIIB occupancies of the promoter, whereas depleting TBP in kin28-AA strain does so, which bolsters their conclusion that appreciable PIC assembly can occur in the absence of detectable intact mediator, and thus suggesting that intact Mediator is not an obligate stoichiometric component of the PIC. There are however a few issues that should be addressed:*

*1) In the experiment of Figure 4, they show that co-depleting Med17 and Kin28 does not reduce PIC assembly and transcription beyond the reduced levels conferred by depleting Kin28 alone, despite reductions of other Mediator subunits to background levels, and in Figure 4—figure supplement 3 they now show that depleting TBP in the kin28-AA strain evokes a much stronger reduction in PIC assembly and transcription, which together provide key evidence for their conclusion that Mediator is not a required component of the PIC, despite its established role in stimulating PIC assembly. Given that the tail domain is still recruited in the absence of Med17; and given their results from the triple mediator mutant implying that on depletion of Med17 alone either (i) mediator tail provides some function(s) in PIC assembly, or (ii) a functionally relevant albeit undetectable low level of intact mediator is still recruited, it seems important to state their conclusion about Mediator being dispensable for PIC assembly at bit more clearly. I think that they are justified in stating only that intact Mediator is not required at detectable levels at the UAS or promoter for appreciable PIC assembly; and to admit the possibility that the residual PIC assembly detected in Figure 4, and also inferred to exist by the downward shift of Pol II in Figure 3, could be dependent on the Mediator tail domain that is still recruited on Med17 depletion, or is dependent on a low level of intact Mediator that might still be recruited in the single med17-AA background that functions catalytically rather than as a stoichiometric PIC component. I feel that this more nuanced interpretation should appear in the Discussion section.*

The Reviewer’s point here is very well taken. In fact, we now realize that part of the confusion was poor wording in that we sometimes used the term “Mediator” to refer to the complete complex and other times to certain domains or subunits. So, we have tried to clarify the confusion about this. The Reviewer is correct that the substantial transcription seen upon Med17 depletion is tail dependent. As such, the tail stimulates PIC formation. However, as discussed in comment 1C of the new experiment section, it is highly unlikely that the tail domain is part of the PIC or even recruits the remaining head/middle subunits to an appreciable extent. So, the vast majority of transcription observed upon Med17 depletion has to be from a Mediator-lacking PIC (it is likely that a small amount comes from incomplete depletion). I hope this Reviewer likes the new Discussion.

*2) Figure 1—figure supplement 1: the Mediator subunit in light blue is missing in the key. It would also be helpful to remind the reader what subunits are in the tail, middle, or head (in the Results section).*

*3) Figure 4—figure supplement 1: the color key is missing.*

*4) Subsection “Pol II transcription can occur from preinitiation complexes lacking Mediator”: cite specific figure panels that allow comparisons of Pol II and/or TFIIB occupancies at the same genes in kin28-AA tbp1-AA double mutant vs kin28-AA med17-AA double mutant that must be inspected to justify the conclusion that depleting TBP has a stronger effect on PIC assembly than does depleting Med17 in the kin28-AA background.*

I don’t know how we can cite specific panels as requested. The relevant figures are cited, and the results are clearly stated, namely considerable GTF occupancy upon depletion of Med17 and Kin28, and virtually no occupancy upon depletion of TBP and Kin28.

*Reviewer #2:*

*The manuscript by Petrenko et al. describes the effect of Mediator subunit depletions by anchor-away on Mediator, RNAPII and GTFs occupancy. Their conclusions can be summarized as follow: The Mediator Core (defined as Head and Middle) and Tail make additive contributions to the role of Mediator in transcription. None of these two functions is essential on its own, but their additive disruption leads to the inability to transcribe. Mediator is therefore described as essential but through two non-essential functions. This contrasts with the current view of an essential Core and a dispensable Tail. In addition, the work provides additional insights such as i) a role for Mediator in the inhibition of promoter escape and ii) in vivo support to the stimulation of TFIIH kinase by Mediator.*

As discussed in the new experiment section, the “missing logic” argument and new experiment 1 (new Figure 5) invalidate the model that Mediator is an obligate component of the PIC. As we now explicitly show, and was predicted from all current knowledge, incomplete depletion of a true GTF reduces transcription but does not alter the relative GTF occupancies, because the structure of the PIC is the same in both cases. In contrast, Med17 depletion drastically alters the Mediator:GTF occupancy/transcription ratio. I note again that this is the exact same logic that allowed us (and Michael Green independently) to demonstrate PICs that do or do not contain the TAF subunits of TFIID.

*The experiments are well executed, and care is taken trying to control for everything but the study is suffering from the inherent limitation of the anchor-away technique: incomplete depletion cannot be ruled out. In addition, a weakness resides in the fact that key conclusions are based on negative ChIP data (i.e. failure to detect Mediator at core promoters is interpreted as a complete absence of the complex). Yet, failure to detect a protein by ChIP could always be interpreted as presence but under the detection limit. This is particularly relevant for Mediator given the inherent transitory nature of the interactions it makes with UAS and (especially) promoter regions. Previous work by the authors represents a good demonstration of that concept: in WT cells, Mediator is not detectable at promoters, yet it is now though to be there, but too transiently to be detected by ChIP. For the conclusions stated by the authors to stand, one would have to rule out the following (and perhaps other) scenario:*

Regarding the issue of a Mediator-lacking PIC, the Reviewer appears to misunderstand our previous paper on Kin28 and Mediator (Wong et al., 2014). In that (and the present) paper, the wild-type PIC is defined as the physical entity that contains GTFs and Mediator, in which the occupancy ratios of these factors are constant at all core promoters and linked to transcription. The PIC can only be detected in a Kin28-depletion strain, although it presumably exists in a wt strain. In a wt strain, the Mediator-containing PIC is very unstable due to Kin28-dependent Mediator dissociation. Upon dissociation, the remaining GTFs are either present as a post-escape complex and/or a Mediator-lacking PIC. In our 2014 paper, we described this as a post-escape complex unable to mediate transcription based on the assumption that Mediator was a general transcription factor. In that 2014 paper, we had no way to determine if there was any Mediator-lacking PIC; that is the value of the new experiments in this paper. In the experiments here, Med17 depletion results in no detectable Mediator at the core promoter even under conditions where a standard PIC is stabilized (Kin28 depletion). This Mediator-lacking entity is thus either a Mediator-lacking PIC or a post-escape complex, and the latter, by definition, does not support transcription, which of course is clearly observed.

*Core Mediator (Head and Middle) is essential for transcription and is an essential component of the PIC (the alternative model to the one proposed by the authors) but the observations presented in the current manuscript are explained by the following: Depletion of Mediator subunits is incomplete (as acknowledged by the authors) and minutes amount of intact core Mediator and/or remaining core Mediator sub-complexes are able to nucleate enough PICs to sustain reasonable amount of transcription; aided by the disconnected Tail bound at enhancers. When, in addition, the Tail is removed, the remaining parts no longer manage to sustain PIC assembly. The arguments in the manuscript against this scenario reside in the fact that Mediator occupancy at core promoters (as measured in kin28-AA conditions and shown in Figure 4) is decreased to background levels upon med17-AA. While this illustrates the important contribution of Med17 on Mediator integrity and function, it does not rule out the possibility that Mediator or Mediator parts make transient contacts with the PIC.*

We agree that we were somewhat careless on our use of the term Mediator (see comment 1 to Reviewer 1) and hopefully have fixed this. As discussed in a section above on the tail module and apparent paradox, we agree that the tail module plays an important transcriptional role upon Med17 depletion. Under these conditions, the tail is not detected at the core promoter and hence is not part of the PIC, but it certainly could help PIC formation either by directly contacting a GTF and/or by having an indirect effect via chromatin or SAGA that helps recruit the PIC.

*The model proposed by the authors is also inconsistent with the observation that most Head and many Middle subunits are essential. The fact that anchoring-away these essential subunits does not prevent growth is the proof that depletion is incomplete. Otherwise, how could these cells grow? The proposition that the essential role of these proteins resides in the cytoplasm is not supported by evidence. The authors argue that because anchoring away GTFs leads to loss of growth, the ability to grow after depletion of essential Mediator subunits is unlikely to be due to incomplete depletion. It appears likely to this reviewer that completely removing Mediator would be more difficult than depleting GTFs. GTFs are, for most of them, single subunit or small complexes. Mediator, on the other hand, is made of 25 different polypeptides. Also, it is not very difficult to imagine that higher amounts of GTFs than Mediator are required for transcription. GTFs are stable components of the PIC and many may even stay in place after escape as a scaffold. This is very different from Mediator, which very transiently associates with the PIC, as the authors themselves showed in a previous publication.*

Our model is not at all inconsistent with the growth experiments. It is incorrect to assume that lethality observed upon deletion of an essential Mediator subunit is due to a complete loss of Pol II transcription. Obviously, the growth observed when an essential subunit is depleted reflects incomplete depletion of that subunit. Under this condition, the observed transcription is mediated by both 1) a Mediator-lacking PIC whose existence is clearly demonstrated by the occupancy ratios, and 2) standard transcription from a Mediator-containing PIC that exists because of incomplete depletion. Although it is impossible to analyze transcription in a strain deleted for an essential Mediator subunit, we presume the Mediator-lacking PIC can still support transcription, but transcription due to incomplete depletion is eliminated. This lower (but still extant) level of Pol II transcription in the deletion vs. depletion condition could be the difference between death and life. This life/death difference could be due to overall reduced transcription or reduced transcription of 1 or more specific genes. Although not often appreciated, the life/death line in yeast often involves small differences in function (I could provide many examples). In this regard, the viability of Mediator-depletion strains is on the margin, as evidenced by slow growth at 30^o^C and essentially no growth at 37^o^C (this phenotypic profile is typical for cells on the margin of life).

Reviewer 2 has speculated about why depletion of Mediator and GTFs might be different with respect to incomplete depletion and the level of protein required for transcription. These speculations might be correct, although there is no actual evidence. On the contrary, the fold-reduction of protein levels upon anchor-away are comparable between Mediator subunits and GTFs (and all other factors we have analyzed) and this includes Pol II and other multiprotein complexes, so we can speculate that the efficiency of anchor-away is comparable for different proteins. All of this is speculation without evidence, and these speculations are unimportant (see point 5). In addition, as we now show, anchor away of a head and middle subunit (both essential) results in cell growth and substantial transcription, whereas anchor away of an essential head subunit and 2 non-essential tail subunits eliminates growth (new Figure 8) and transcription (new Figure 7). This module specificity is not easy to explain by incomplete depletion.

Although consistent with our conclusions, the growth experiments *per se* do not represent the key evidence for these conclusions. Instead, our conclusion about a transcriptionally competent Mediator-lacking PIC is based on the factor occupancy ratios and the downstream shift. As such, the growth experiments do not significantly affect the key conclusions of the manuscript.

*This reviewer is also not convinced about the shift in Pol II distribution shown in Figure 3. This effect is very small and it is not clear that it can be interpreted as a faster escape. It looks like the curves have been re-scale so that they all reach the same level at the 0.5kb position (please confirm). This alone may introduce distortions in traces like the one depicted here. Also, given that Mediator stimulates TFIIH-mediated phosphorylation of the CTD, the interpretation made by the authors here is very counter-intuitive.*

The comments about the Pol II shift are incorrect, and this was addressed in the last round of review. Reviewer 2 may think the effect is “small” (whatever that means), but it is qualitatively obvious upon inspection, statistically significant, and specific to head/middle vs. tail/kinase module which is all that matters. It is worth noting that the curves are based on the combined data of hundreds of genes, which makes the results robust. Fundamentally, the Pol II profile is defined by the relative Pol II levels at various positions along the gene. As such, the scaling doesn’t matter, because it doesn’t affect relative Pol II levels along the gene. We scaled everything to the same level at the 0.5 kb position to make the data more understandable, but the scaling *per se* doesn’t affect anything. I don’t really understand why Reviewer 2 thinks are conclusion is counterintuitive because Mediator stimulates TFIIH activity. Yes, in this respect Mediator might act in the same manner as Kin28, but the stimulation is only partial and is highly likely to be outweighed by the previously shown inhibitory effect of Mediator on promoter escape. As nicely and recently shown by the Malik paper that was pointed out to me by Reviewer 3, the functional relationship between Mediator and TFIIH is complicated.

In sum, the model proposed by the authors may very well be right but alternative explanations are just as likely to be right. The data presented in this manuscript has value but the interpretation that is made of it goes too far.

As mentioned in the last rebuttal letter, Grunberg et al., 2016 does show some Mediator occupancy at ribosomal and glycolytic enhancers, but this level is far below the level seen at other activated genes and is not correlated with transcriptional activity. In this respect, Grunberg et al. and our previous work are in agreement, except perhaps in terms of assay sensitivity. Also, Grunberg et al. doesn’t see increased Mediator association upon Kin28 depletion or inactivation, which was seen in 2 different labs. We have cited Grunberg et al. in a fair manner, so I’m not sure what the issue is.

As now discussed here and in the paper (the missing logic argument), transcription due incomplete depletion should have all the properties of transcription prior to depletion, because the same protein is involved, just less of it. As such, the fact that there is a downstream shift means that the PIC that mediates this transcriptional profile can’t be the normal PIC, but rather a PIC that lacks Mediator.

We have corrected the misleading and poorly written sentences saying that Mediator is not required for PIC formation. It IS required for PIC formation, but is not an obligate component of the PIC. This is the interesting and unexpected conclusion.

*Reviewer #3:*

*[…] In summary, I do think the findings are of broad interest and well documented. They shed considerable light on old data regarding transcription occurring independently of (actually in med17-ts yeast) Mediator, and provide new insight into the role of Mediator in PIC formation (needed only transiently, perhaps not at all) and transcription. A more rigorous interpretation of results would, in my opinion, enhance impact.*

The major comments of Reviewer 3 are addressed in the preceding comments. In particular, we agree that Mediator is not required for PIC formation and have corrected the sloppy statements in this regard. We did not intend to make this conclusion (it is obvious that the Mediator requirement for transcription means that it is also required for PIC formation, since PIC formation is required for transcription. What we meant to say (and said on multiple occasions) is that Mediator is not an obligate component of the PIC, and this conclusion was *not* made in our previous paper; quite the contrary (see point 2 of response to Reviewer 2).

[Editors’ note: the author responses to the first round of peer review follow.]

*[…] Finally, the authors may need to reinterpret their data on SAGA- versus TFIID-dependent genes, and Mediator occupancies at RP and glycolytic genes, in light of other relevant publications.*

The amount of additional work being requested to address the reviewers' criticisms is considerable, and it is unclear whether the results of these new experiments would support the authors' main conclusions. Thus, it is impossible to judge the work as potentially acceptable with suitable revision. However, this same group of referees would be willing to consider an extensively revised manuscript containing significant additional experimentation and analyses that would address all of their serious concerns.

New Experiments:

1) As requested by Reviewer 1, we now document our re-analysis of the published Morse genome-wide data on the *med17*-ts mutant (new Figure 1—figure supplement 1). In complete accord with our Med17 anchor-away results, there is a general, but modest effect on transcription. We also include IGB screen shots of Pol II occupancy at several genes in the Morse vs. our data (new Figure 1—figure supplement 2). This not only shows the similarity between the 2 datasets, but also that the heat shock genes are up-regulated upon temperature shift in the ts strain despite inactivation of Med17; this does not occur in the anchor away strain, because there is no temperature shift.

2) To improve on point 1 even more, we performed our own experiments comparing Med17 ts and anchor away strains; analysis of 8 genes by qPCR analysis and got similar results (new Figure 1—figure supplement 3).

3) As requested, we have performed genome-wide occupancy of Pol II in the triple mutant (new Figure 6). As expected, transcription is severely reduced to a level indistinguishable from that seen upon depletion of TBP.

4) As requested by all the Reviewers, we performed statistical analysis on the Pol II shift experiments. In addition, we analyzed new biological replicates (whole-genome scale). As was qualitatively clear from the graphs, the downstream shifts observed in Med22, Med14, and Med7 depletion strains are clearly significant, whereas no downstream shift is observed in Med15 and Cdk8 depletion strains. The Med17 is also significant when averaging the replicates, but somewhat ambiguous in that one replicate shows the downstream shift, whereas the other does not. It is not straightforward to calculate *p-*values because the curves consist of Pol II occupancy values at multiple positions, and the values at these positions are not completely independent of each other. Therefore, we calculated the *p-*value for occupancy values at +100, and these are presented in the methods. The overall significance is much stronger than these reported *p-*values, because they do not include differences that are apparent at other locations (Figure 3; -100 to +300). Importantly, the downstream shift is qualitatively obvious upon inspection, and it is clearly documented in multiple head/middle subunits in contrast to the wt strain and all tail mutant strains tested; hence the conclusion is valid. Moreover, this result is not surprising given previous results that Mediator inhibits promoter escape (Kin28 depletion) and the recent in vitro work of Malik (see comment 4 of Reviewer 3).

5) As requested by Reviewer 1, we analyzed transcription in a strain simultaneously depleted for Kin28 and TPBP. As expected, we do not detect occupancy of Pol II, TFIIB, TBP, and Mediator at the core promoter (Figure 4—figure supplement 3).

6) As requested by all Reviewers, we demonstrate that in strains depleted for Med17 ± Kin28, TBP and TFIIB occupancies at the core promoter parallel that of Pol II, not Mediator (Figure 4 and Figure 4—figure supplement 1 and Figure 4—figure supplement 2). The comparable levels of TBP, TFIIB, and Pol II at the core promoter in the virtual absence of Mediator conclusively demonstrate our main conclusion or a Mediator-lacking PIC that supports transcription. Importantly, consistent results are observed for all 7 genes tested, indicating the generality of the conclusion.

7) To improve the genome-scale transcriptional analysis (Figure 1), we analyzed a second general transcription factor (Pol II), and we performed replicates on all the strains; these replicates are very highly correlated.

General comment:

The title starts with “Evidence that…”. We recognize and strongly emphasize that it is impossible to do a perfect experiment that involves complete removal of an essential protein. So, right in the title, we qualify the key conclusions and do not make them definitive, which is impossible. Nevertheless, it is important to note that some of the critical conclusions depend on novel transcriptional patterns (e.g. downstream shift and low Mediator:Pol II/GTF ratio at the core promoter) that cannot be explained simply by incomplete depletion.

*Reviewer #1:*

*[…] General critique:*

*The conclusion that Mediator is essential for transcription in vivo, based on the results of simultaneous deletion/elimination of three Mediator subunits seems well demonstrated by the results in Figure 6. These data represent a more definitive finding than the original results of Young et al. that were based on mRNA measurements alone and did not rule out secondary effects on mRNA stability; and which are apparently in conflict with more modest effects on transcription observed by the authors in re-analyzing published data from the Morse lab. It seems necessary for the authors to document their re-analysis of the Morse data to justify this claim. Moreover, their interpretation of the additive effect of combining head and tail deletions as evidence for distinct functions of these modules is not compelling as it overlooks the possibility that the removal of Med17 from promoters is simply more complete in the triple mutant owing to the loss of the tail subunits and their known function in Mediator recruitment by activator proteins. The authors acknowledge that anchor-away of Med17 is unlikely to be complete, and it has been demonstrated previously that tail subunit deletions reduce the occupancy of head/middle subunits, at least at certain promoters (see specific comments below for details). As such, it is entirely possible that there is a functionally significant level of Med17 recruitment, albeit below the detection limit of their ChIP assays, that is retained in the med17-AA single mutant but reduced further or even eliminated in the triple mutant owing to loss of the established role of tail subunits in Mediator recruitment to enhancers. It seems that this interpretation should be included in the Discussion as an alternative to their suggestion that the tail has a distinct function beyond Mediator recruitment, for which they cite no other evidence.*

*The second important conclusion, that PIC assembly can proceed without Mediator, is a plausible interpretation of the results in Figure 3, which imply enhanced Pol II release from PICs assembled at low levels of Med17. However, it is important to establish the statistical significance of this shift. The supporting results in Figure 4 are more complicated, however, because the kin28-AA mutation reduces transcription dramatically on its own, making it difficult to determine whether or not co-depletion of Med17 could evoke a further decline that could be measured at four of the five genes they analyzed; and it seems necessary to establish that co-depletion of a GTF (e.g. TBP) would reduce transcription further in the kin28-AA cells, in contrast to depleting Med17. In addition, it's difficult to understand why co-depletion of Med17 would not reduce PIC assembly whatsoever even if Mediator is not essential for this process unless Mediator is completely uninvolved in recruiting Pol II, which seems unlikely. It is worth noting that it was shown previously that TBP recruitment to Gcn4 activated promoters is impaired in single mutants lacking either a head or tail subunit (Qiu, H., et al. (2004). Mol Cell Biol 24(10): 4104-4117.); and there may be similar such measurements published for other activated genes. Hence, to bolster their conclusion that Mediator is not a required component of the PIC, it is important that they attempt to demonstrate this point more directly by analyzing TBP or TFIIB promoter occupancies on Med17 depletion, in both WT cells and in the double mutant lacking the tail subunits Med3 and Med15. If their thesis is correct, appreciable TBP/TFIIB recruitment should be maintained in the absence of these Mediator subunits.*

*Finally, it's unclear whether they believe that Mediator has no role in PIC formation or that it is simply not required as a stable, stoichiometric constituent of the PIC. If it is not required for PIC formation, then what is its essential role in transcription? Is it enhancing Kin28 function to allow promoter escape? Is that essential? The paper should discuss what the essential role of Mediator could be if it is not to stimulate PIC assembly.*

*Specific comments:*

Introduction section: the re-analysis of the Pol II ChIP-chip data published by Paul et al. should be presented as a detailed supplementary figure to substantiate this claim.

*Results section: the reduction in transcription at PMA1 on depletion of Med17 is not modest, so this claim should be qualified to describe the average behavior.*

*Results section and Figure 1: PMA1 is the only gene for which both Mediator and Pol II occupancies were measured, and both occupancies are dramatically reduced by Med17 depletion. It seems important to examine Med17 occupancies of all of the genes whose Pol II levels were examined in this figure, rather than simply assuming that every gene exhibits the same strong reduction shown for CCW12 and SED1.*

Figure 1 and Figure 2: it seems important to acknowledge that the AA-tags introduced into Mediator subunits reduce their functions appreciably in the absence of rapamycin.

Figure 2: the color-coding should be labeled as log_2_ [(Pol II(-rap)/Pol II(+rap)] or something similar.

*Figure 3: it is important to establish the statistical significance of the shift in Pol II downstream.*

*Subsection “Pol II transcription can occur from preinitiation complexes lacking Mediator” and Figure 4: the interpretation of these data is complicated by the fact that transcription is reduced extensively at four of the genes by depletion of Kin28 alone, making it difficult to determine whether a further reduction ensues with Med17 depletion. Would they be able to see a further reduction if they depleted a factor essential for PIC assembly, e.g. TBP, in the kin28-AA mutant? This control would seem to be essential to support their interpretation. In addition, even if Mediator is not essential for PIC assembly, which seems reasonable to conclude for CUP1, wouldn't they expect that Med17 depletion would reduce PIC assembly at this gene by decreasing Pol II recruitment? One way of interpreting these data is to conclude that Med17's role in stimulating transcription at these genes is completely dependent on Kin28, which in turn could mean that Med17 acts primarily by enhancing Kin28 function in promoter escape (kin28-AA is epistatic to med17-AA)? It also seems difficult to eliminate a role for Mediator in PIC assembly where it would not have to function as a stable, stoichiometric component of the PIC, i.e. more like an enzyme. In fact, significant Med17 occupancy is detected at HSP82 and CUP1 following depletion of Med17 in the kin28-AA strain-perhaps this low level is sufficient.*

Subsection “Pol II transcription is virtually eliminated when Mediator head, middle, and tail modules are simultaneously inactivated” second paragraph: shouldn't this read: "…in the double mutant strain… (Figure 6)"?

*“Thus, depletion of all three Mediator modules has a stronger transcriptional effect than conditions where the tail module is present at enhancers (Med17 depletion) or the head and middle modules are present at core promoters (deletion of tail subunits)”: this statement should be revised as it seems to imply that deletion of the tail subunits does not reduce Med17 occupancy, whereas this is very unlikely, based on previous studies showing a strong reduction in Head subunits when the tail is deleted (e.g. the reference Zhang et al. (2004)). Also tail subunit occupancies could be reduced, even if not eliminated, by depletion of Med17.*

*“As deletion of some Mediator subunits […] grow at 37C (Figure 7)”: there are several parts of this sentence that refer to published findings that are not cited.*

*Subsection “Growth of Mediator-depletion strains” paragraphs two and three: it seems possible that depletion of Mediator subunits is less deleterious than depleting GTFs because Mediator can perform one or more functions catalytically at much lower cellular levels without being a stable, stoichiometric constituent of the PIC, e.g. stimulating Kin28 kinase activity.*

1) As described in point 1 of “new experiments”, we confirm that of loss of Med17 function via the ts mutant or anchor-away causes a general, but modest effect on Pol II transcription. Thus, our results supporting the conclusion that Mediator is essential for Pol II transcription are not merely “more definitive” than all previous results, but rather they give a different answer. Specifically, we demonstrate that Mediator is required for Pol II transcription in vivo. All previous papers (e.g. Morse) show only a partial loss of Pol II transcription, clearly distinct from what happens upon inactivation of Pol II or TBP, which at face value would indicate that Mediator is not essential for transcription. In this sense, our conclusion is novel.

2) We agree with the Reviewer that our conclusion of “distinct functions” of the tail and head/middle modules in transcription is misleading and perhaps overstated. We have modified/amplified the text in this regard and removed this conclusion from the Abstract. Reviewer 1 is correct that the “function” of the tail module in strains depleted of Mediator head/middle subunits could be simply to recruit the limiting amount of Mediator to the core promoter; i.e., no special function of the tail module beyond Mediator recruitment. Of course, recruitment to the core promoter is a different Mediator “function” than activity at the core promoter, but Reviewer 1 is correct to point out we implied more in our original manuscript.

3) Related to point 2, the issue of whether the tail module functions solely in recruitment of Mediator to the core promoter is extremely difficult to address directly. In yeast, where transcription activity is highly correlated with GTF occupancy, any transcriptional role of the tail module is highly likely to affect PIC levels, and Mediator is part of the PIC in wild-type strains. Thus, loss of the tail module is predicted to cause reduced Mediator association at the core promoter under virtually any circumstance. As a consequence, the comparison of the triple mutant with the individual Mediator subunit depletions does not directly address the question of whether Mediator is essential for PIC function, our second major conclusion. In addition, while this particular result is consistent with our conclusion that Mediator is not essential for PIC function, it provides only weak supportive evidence. Importantly, however, our conclusion about Mediator being dispensable for PIC function is based on several other independent and more convincing experiments (see below).

4) Related to point 3, I think it unlikely that the general transcriptional role of the tail domain (i.e. the general reduction in Pol II transcription seen in tail mutants ± Med17 deletion) is due solely to recruitment of Mediator to the core promoter. In particular, Mediator occupancy at the enhancer is poorly correlated with transcription, and indeed many genes (e.g. ribosomal protein genes) have very high levels of transcription with minimal Mediator occupancy at the enhancer. Nevertheless, as stated in point 2, we have downplayed the “distinct functions”.

5) We are pleased that Reviewer 1 recognizes that the downstream shift in Pol II profiles upon head/middle, but not tail depletion provides strong support that a functional PIC can be generated without Mediator. The results are highly unlikely to be due to a “scaling artifact”. The scaling method is very standard, internally controlled within each sample, and the same scaling method was used in our previous work on Kin28 (Wong et al., 2014). It seems extremely unlikely that, by chance, the head and middle subunits behave in a similar way to generate a downstream shift, the tail subunits have no effect, and Kin28 has the opposite upstream shift. As stated in point 4 of new experiments, our statistical analysis demonstrates that the altered pattern is highly significant (although the Med17 result is less clear).

6) The low Mediator:Pol II ratio observed in strains depleted of both a Mediator subunit and Kin28 represents our most direct evidence supporting the conclusion that Mediator is not a required component of the PIC. The request of Reviewer 1 to analyze cells simultaneously depleted of Kin28 and TBP is based on the incorrect premise that “Kin28 depletion reduces transcription dramatically”. In fact, several publications including ours indicate that depletion of Kin28 has only a modest (~2-fold) effect on transcription, so it is very easy to see a further decrease in transcription. Nevertheless, we did perform the suggested TBP/Kin28 double depletion, and obtained the expected result of essentially no transcription (point 5 in new experiment section).

7) Related to point 6 and of more importance, we performed the requested experiment of analyzing TBP and TFIIB occupancy at core promoters under conditions where Med17 and Kin28 are simultaneously depleted (see point 6 in new experiments). As expected, TBP and TFIIB levels are consistent with Pol II levels, indicating that the Mediator:TBP and Mediator:TFIIB occupancy ratios are also reduced. This clearly shows that it is possible to obtain a functional PIC (contains TBP and TFIIB and supports Pol II transcription) with very little Mediator present at the core promoter. We do not understand why simultaneous depletion of Kin28 and Med17 does not reduce Pol II transcriptional significantly more than individual depletion of these proteins. It could be that the opposing effects on Pol II escape (Kin28 depletion inhibits escape and Mediator depletion facilitates escape) cancel each other out (also seen in Malik paper), but we really don’t know. Nevertheless, this issue does not affect the measurements of Pol II:GTF ratios, which is the basis of the critical conclusion that Mediator is not essential for PIC formation/function.

8) We do not believe that Mediator has “no role in PIC formation/function”. Quite the contrary, we specifically state that Mediator stimulates PIC formation, and now include this in the title of the paper. Our conclusion that Mediator is not essential for PIC formation/function is not inconsistent at all with stimulation of PIC formation; they are not mutually exclusive. We do discuss the issue of the essential role of Mediator and specifically note that there may be multiple functions besides forming a functional PIC. Even in the simplest case, recruitment of the PIC is different than forming a functional PIC.

*Reviewer #2:*

*In this manuscript, Jin, Struhl, and colleagues report "Evidence that Mediator is essential for Pol II transcription but not a required component of the preinitiation complex in vivo". Although the results presented are interesting, I don't believe they are conclusive with regard to the strong statements made in the title and Abstract.*

*Much evidence exists in the literature for the importance of Mediator in mRNA transcription. Thus, the distinction between Mediator being important and essential is critical for the impact of this manuscript. The evidence that Mediator is essential for Pol II transcription is based on the "triple" strain, med3∆ med15∆ med17-AA, in which Med17 is depleted from the nucleus using the anchor away technology. Figure 6 shows that there is indeed a strong effect on Pol II occupancy in this strain. However, the effect is not complete, as the authors acknowledge. In fact Pol II is still enriched at CUP1 by about 15 fold, or about four fold less than in WT yeast, and close to ten-fold at HSP82, about a five-fold reduction from wild type levels. The authors interpret this as being more likely to be caused by incomplete Mediator depletion than by activation without Mediator by comparing the reduction of Pol II occupancy after depleting Rbp1 or TBP using anchor away. Although they state that the effects are "roughly comparable to (although perhaps slightly lower than) that occurring in the triple deletion strain", it seems clear that the effect really is less. The graphs for the triple deletion strain ought to be presented side by side with the rpb1-aa and tbp-aa strains; it's not clear that "perhaps" applies, and therefore the argument that Mediator is truly essential-that transcription absolutely depends on its presence-is weakened. In addition, given the importance of this result, Pol II occupancy ought to be measured genome-wide as it was for individual anchor-away experiments in earlier figures.*

1) I think our statement that the very low level of transcription in the triple mutant is roughly comparable to (although perhaps slightly less pronounced) than observed upon depletion of TBP or Pol II is an accurate description of the results. In all cases, it is impossible to completely eliminate the essential protein, so some transcription remains. They key here is not the level *per se*, but rather the comparison between the triple mutant and known GTF controls. Our genome-scale comparison of transcription in the triple mutant and TBP depletion (point 3 of new experiments) yields indistinguishable results. While we can’t definitively prove that Mediator is 100% essential for transcription (that is impossible), we do show that any Mediator-independent transcription is very low at best. This is a very different conclusion from anything previously done, where substantial transcription remains when Med17 is inactivated by the ts mutant.

*The second principal conclusion of the paper is that Mediator is not a required component of the PIC in vivo. This is based on experiments in which depletion of Med17 by anchor away, or inactivation in the classical med17-ts mutant, results in only partial loss of Pol II association with ORF regions while Mediator occupancy at promoters (seen by also anchoring away Kin28) is greatly reduced. Reduction of Mediator occupancy is only shown for three FRB-tagged Mediator subunits at the CCW12 enhancer and for med17-aa at three promoters. Depletion should be examined genome-wide for at least some of the anchor-away strains, probably best while also depleting Kin28. In addition, the authors emphasize that the low Mediator:Pol II occupancy ratio at the core promoter seen when both Med17 and Kin28 are depleted "provides very strong evidence of Mediator-independent PIC formation and function". But measurement of this same ratio in KIN28+ yeast would also yield a very low Mediator:Pol II ratio; is it not therefore possible that dynamics still play a role in this measurement and that it does not provide a completely accurate picture of PIC composition in vivo? In addition, once Pol II escapes the promoter, it will continue to contribute to ChIP occupancy measurements but will no longer be part of a PIC as normally understood. It would be more convincing to also measure occupancy of other PIC components such as TBP or TFIIB in this experiment and compare them to Mediator occupancy; but even if this were done, questions of dynamics with regard to Mediator would persist.*

2) In the original version of the paper, the key conclusion that Mediator is not required for PIC formation/function relied primarily on measurements of the Mediator:Pol II occupancy ratio under conditions of Med17 and Kin28 depletion (Figure 4). As Reviewer 2 points out, we measured Pol II occupancy at the coding region, not the core promoter, and the altered behavior of Pol II promoter escape could potentially influence the results and hence the conclusion. To address this criticism and as requested, we now analyze TBP and TFIIB occupancy at the core promoter (point 6 of new experiments) and show that the Mediator:GTF ratio is drastically reduced. This clearly demonstrates that a functional PIC can be generated with very low levels of Mediator at the core promoter.

3) I disagree with the comments about the Mediator:Pol II/GTF ratio and Mediator dynamics and think the experiment in Figure 4 (and supplements) very strongly supports our key conclusion that Mediator is not required for PIC formation/function. The dynamic behavior that distinguishes Mediator from any other transcription factor is not related to Mediator structure *per se*, but rather to Pol II CTD phosphorylation. For the discussion below, we define the PIC in wt cells as a Mediator-containing entity, as shown in our (and Francois Robert’s) previous paper and in accord with what everyone thinks.

In wt cells, the PIC is transient because, upon formation, the Pol II CTD is rapidly phosphorylated by Kin28 whereupon it dissociates from the promoter. As a consequence, we can’t actually detect Mediator in wt conditions, and hence the Mediator:Pol II ratio is a useless concept. In addition, it is very clear from several publications that Mediator association at enhancers is very poorly correlated with transcription. This high variability in Mediator association at enhancers has nothing to do with dynamic differences related to CTD phosphorylation, but rather more simply on the efficiency of activator-dependent recruitment.

Upon Kin28 depletion, we see very high Mediator at the promoter and this represents the PIC. THE KEY POINT is that the level of Mediator in this condition is highly correlated to the level of Pol II transcription (shown in our previous paper and the companion paper from Francois Robert) as well as TBP and TFIIB occupancy (our new data). So, under these Kin28-depletion conditions, a given level of Mediator-containing PIC gives rise to a predicted level of transcription just in the same way that a general factor does. There is no reason to believe that the intrinsic ability of Kin28 to phosphorylate the CTD and cause Mediator dissociation is promoter dependent; quite the contrary.

As such, in wt cells, we presume that the Mediator:Pol II:TBP:TFIIB ratio is constant at all promoters, but that we simply can’t measure Mediator because it is transient under these conditions; i.e. the PIC contains Mediator, but we can’t measure it. In yeast, the level of PIC formation determines the level of transcription (there is no promoter-proximal pausing as occurs in flies and mammals).

So, under Kin28-depletion conditions, the level of the PIC (which means Mediator occupancy or any general subunit) yields a certain level of transcription. However, when we simultaneously deplete Kin28 and Med17, the Mediator:GTF ratios decrease dramatically, but the Pol II:TFIIB and Pol II:TBP ratios do not and neither does transcription. Unlike the drastic difference in Mediator:GTF occupancy ratios in wt vs. kin28 mutant cells that depends on an actual change in Mediator interaction with Pol II and not transcription, the difference in kin28/med17 vs. kin28 cells depends only on the level of Mediator under conditions where PIC levels are strictly correlated with transcription. This means that when Mediator is depleted and the Mediator:Pol II/GTF ratio goes down, the observed transcription must come from a Mediator-lacking PIC. This is exactly the same logic we and Michael Green used years ago to demonstrate TAF-lacking PICs and TAF-independent transcription.

*Another piece of evidence cited for Mediator not being required for PIC formation in vivo is the downstream shift of Pol II seen upon depletion of Mediator head or middle subunits. However, this shift (Figure 3) is observed most strongly for med7-aa yeast, somewhat less strongly for med14-aa, and not at all for med17-aa. Thus, the evidence for this downstream shift of Pol II is ambiguous. Also, a downstream shift in Pol II was reported upon Kin28 inactivation by Rodriguez-Molina et al. (2016) Mol. Cell 63:433, and interpreted as representing a yeast-specific elongation checkpoint. The authors should discuss this result in light of their own findings.*

The downstream shifts observed in Med22-, Med7-, Med14-, and Med17-depleted cells are statistically significant (see point 4 in new experiments). It is true that the degree of shift differs among these strains (for unknown reasons), but the shifts *per se* are strong evidence for an effect on promoter escape. Reviewer 2 is incorrect about the Rodriguez-Molina paper, which did not show a downstream shift upon Kin28 inactivation, but rather increased accumulation of Pol II at the +2 nucleosome and inhibition of escape into elongation. This latter point of decreased escape into elongation is what we observed as well (although we analyzed the promoter, not the +2 nucleosome). I also note that the upstream shift in Kin28 mutants was not only observed in our 2014 paper, but also in earlier data from the Robert lab.

*A third major conclusion is that Mediator modules make independent contributions to the overall transcriptional function of Mediator (Discussion section). If I understand the argument correctly, the authors suggest that in med17-aa yeast, association of the tail module of Mediator with enhancers is sufficient to activate many genes in the absence of middle/head module function. Since there is no evidence for such independent function (albeit independent recruitment has been demonstrated) of the tail module, it seems more economical to postulate that decreased function of the middle/head module, or continued function of the middle module (which does make contact with PIC components), accounts for remaining activity. The authors also appear to argue that the "apparent absence of Mediator at enhancers that drive expression of ribosomal protein and glycolytic genes" could be due to the middle/head modules functioning independently of the tail module at these genes. But low Mediator signal at these signals actually appears to be caused by the same Kin28-Pol II CTD mediated dynamics that the authors and the Robert group have reported, and that is used here to advantage. For example, see Jeronimo and Robert (2014) Figure 3D and Supplementary Figure 2B.*

As for independent contributions of the head/middle and tail domains, please see responses 2-4 to Reviewer 1.

*Reviewer #3:*

*The manuscript by Jin et al. provides two fundamental conclusions about Mediator in yeast cells: 1) Mediator is essential for transcription in vivo and 2) Mediator is not a required component of the PIC in vivo.*

*The first conclusion was somewhat expected (at least for most people) since a frequently cited study from the Young lab has shown that a ts mutant for Srb4/Med17 leads to massive decrease in steady state mRNA levels. As well articulated by the authors, this old data was not directly addressing transcription so a formal demonstration of the essentiality of Mediator for transcription in vivo was lacking. The authors addressed this question by combining a double deletion of Tail module subunits with the nuclear depletion of the head subunit Med17. This triple mutant was used in Pol II ChIP assays to show decrease in Pol II occupancy comparable to those observed when depleting TBP or Pol II itself. This indeed provides compelling evidence that Mediator is essential for transcription in vivo.*

*The second conclusion, however, is more surprising and also not as decisively supported by the data. In sum, they showed that depletion of individual subunits (Tail, Head or scaffold) -unlike the triple mutant- leads to only partial (2-3 fold) reduction in Pol II occupancy over genes. Because Head and Middle subunits can interact with the PIC in the absence of a Tail, and because the Tail can still be recruited to enhancers in the absence of Head and Middle, they interpret this data to say that Tail and Head/Middle have independent contributions to transcription, none of them being essential on its own. They then looked at Mediator occupancy at promoters in conditions when it is detectable (Kin28-depletion) to show that it decreases to a much larger extent than Pol II upon Med17 depletion. Such a low Mediator:Pol II ratio at promoters in Med17-depleted cells is interpreted as a strong indication that a PIC can form in the absence Mediator in vivo. Although they acknowledge the fact that Med17 depletion is likely to be incomplete, the authors argue that their conclusion can be made independently of a complete depletion. Although I suspect that the authors' conclusion is correct, I think that alternative interpretations cannot be completely excluded. Mediator occupancy at core promoters is very transient. In addition, sub-Mediator complexes have been shown to exist in cells. Could it be that upon depletion of Med17, assembly of Mediator within the PIC still occurs but in a less stable (more transient) manner. This would lead to a decreased Mediator-Pol II ChIP ratio but would not rule out the possibility that this transient interaction is nevertheless necessary for transcription. In essence, Med17 depletion would simply exacerbate a phenomenon already present in WT cells: A transient but necessary interaction of Mediator with the PIC. Again, this is perhaps not the most likely explanation, but I do not think one can rule it out completely. The authors provide evidence to dispute it or if not, acknowledge this possibility and modify the title of their manuscript accordingly.*

Regarding our conclusion that Mediator is not essential for PIC formation/function, Reviewer 3 “suspects that our conclusion is correct” but says that “alternative conclusions cannot be completely excluded”. I agree that we can’t “completely” exclude alternative explanations, but this is true for virtually all scientific publications; unalloyed “truth” is the province of religion, not science. The question is how well these alternative explanations make sense based on current knowledge and how well they fit the data. I should also note that our paper was entitled “Evidence that…” for this reason. So, I think our title adequately expresses what Reviewer 3 is getting at and that we don’t need to change it.

I think the “alternative explanations” mentioned by Reviewer 3 are highly unlikely. First, for reasons detailed in point 3 of Reviewer 2, the transient nature of Mediator occupancy at core promoters is due to Kin28-mediated phosphorylation of the CTD. This is a completely independent phenomenon from the actual function of Mediator at the PIC, and there is no basis for linking these things in the manner suggested.

Second, I’m unaware of in vivo evidence that Mediator sub-complexes exist in wt cells (except for that lacking the kinase module). If the evidence is based on biochemical analysis of Mediator in wild-type extracts, I don’t buy it. Of course, partial complexes exist in mutant cells (e.g. the kinase module is functional in the absence of the head/tail module and *vice versa*). I also note that in our previous paper (Petrenko et al., 2016), we saw loss of head/middle subunits when we individually depleted a head or middle subunit, which is inconsistent with such partial complexes

Third and most importantly Francois Robert and us previously published that Mediator occupancy at the core promoter in kin28 depleted cells is very strongly correlated with Pol II transcription. Reviewer 3 suggests that Pol II occupancy is high in Med17-depleted cells because the residual Mediator associates with the PIC in a more transient manner (and hence not detectable by ChIP). However, this suggestion would necessarily mean that this mutant Mediator (or a much lower level of wt Mediator that remains) would have much higher transcriptional function than wt Mediator at normal physiological levels. Aside from being *ad hoc*, this seems like a remote possibility.

Fourth, the suggestion that “Med17 depletion would simply exacerbate a phenomenon already present in WT cells” is mixing up 2 different situations. As mentioned above, the known transient association of Mediator is linked specifically to Kin28-mediated phosphorylation. The suggestion that the mutant Mediator transiently associates has nothing to do with Kin28, but rather the lack of a core Mediator subunit. These are completely different things.

*Other comments:*

*It is stated in the Introduction and later in the Discussion that Mediator poorly (if at all) associates with the enhancer of RP and glycolytic genes. While this is indeed what is observed by ChIP, a recent paper used ChEC-seq to show that Mediator can be found at virtually all enhancers in yeast, including those upstream of RP and glycolytic genes. This suggests that Mediator detection by ChIP is likely to be dependent on its dwell time on enhancer DNA, similarly to what was shown on promoter DNA. This paper should be cited and the argumentation should be modified accordingly.*

Although the ChEC-seq paper shows some Mediator at many enhancers including RP and glycolytic ones, the level of association is very low. So, the ChEC-seq might be more sensitive than standard ChIP, but the basic point remains; poor association of Mediator. Indeed, the authors of the ChEC-seq paper emphasized that Mediator association at the enhancer is poorly correlated with transcriptional activity, which completely agrees with what we published 10 years ago. The existence of the ChEC-seq paper is why we used the phrase “poorly (if at all)”. I should also point out that ChEC-seq completely missed the strong core promoter association of Mediator in Kin28 mutant strains, so this technique clearly has its own issues. Lastly, all ChIP experiments are time- and cell-averaged measurements. The “dwell time” on enhancers reflects activator-mediated recruitment, and this differs greatly among activators. The “dwell time” at core promoters reflects 2 different states of Pol II. Linking these together is a mistake, as they are completely different mechanistically.

*The data shown in Figure 3 is interesting and in line with a recent study from the Malik lab (PMID: 27916598). This should be cited and perhaps discussed. Although interesting, this analysis is rather thin and not very convincing. Indeed, the shift downstream is very small and required scaling the different datasets. While this is a reasonable thing to do to the data, I am afraid that the shift may be an artifact of the scaling. Can the author further this analysis? Perhaps they could find a group of gene where the effect is more readily visible and show specific examples. Also, can they provide an analysis ruling out the possibility that the shift correlates with the amount of scaling applies to each dataset?*

We thank Reviewer 3 for pointing out the Malik paper, which we were unaware of. Although the Malik experiments are done in vitro, they fit nicely with our results and we have now cited this paper. As for validity, Reviewer 3 has misunderstood the scaling method. The “scaling” analyzes the data in an internally controlled manner. Specifically, it measures the amount of Pol II at various positions in a given sample with respect to overall level of Pol II occupancy in the same sample. We never directly compare Pol II occupancy values between samples; rather the Pol II pattern of a given sample is defined solely by data in that sample. Showing specific examples could be done and will certainly exist, but there would be no way to determine if such examples are within experimental error. Our statistical analysis (point 4 of new experiments) shows that the results are valid, and indeed it is qualitatively obvious that the head/middle subunits behave differently from the tail subunits.

*As mentioned by the authors, the fact that depletion of essential Mediator subunits does not abrogate growth is puzzling. The paragraph on this topic (in the subsection “Growth of Mediator-depletion strains”) is not very compelling. The authors claim that incomplete depletion is unlikely to be the explanation, yet they do not provide any alternative (except for a "non-chromosomal" function, which would be extremely surprising and for which there is absolutely no evidence). Their argument to dismiss incomplete depletion is the fact that depletion of Kin28 is lethal despite a comparable effect on Pol II. This is not a valid argument since it is well established that kin28 lethality is due to defect in post-transcriptional events such as mRNA capping. With the lack of a better explanation, the authors should acknowledge incomplete depletion as the most likely explanation. This also implies, that very little amount of Mediator is necessary for growth, and hence for transcription, which is in line with their ChIP data.*

We agree that the viability of strains depleted for essential Mediator subunits is puzzling, and we don’t really have a definitive answer; the paragraph is not intended to be compelling. We have softened the discussion even further, but do think that incomplete depletion *per se* is less likely. Our reasoning is not just based on the comparison to Kin28, which the Reviewer notes also affects post-transcriptional processes. In this regard, there is considerable evidence that Mediator also affects post-transcriptional processes and perhaps these involve functions when not recruited to DNA. But, the main argument is the striking difference between general transcription factors and Mediator in terms of viability upon depletion by the same anchor-away system. Lastly, our ChIP data does not address this issue, other than we can’t detect the association of any factor upon anchor-away mediated depletion.

*Regarding the Mediator-dependent at SAGA versus TFIID genes, it has recently been argued that this may be due to technical limitations (PMID: 27773677). Can the authors comment on that?*

Our SAGA vs. TFIID results are actually consistent with Jeronimo et al. 2016. That paper said that the results were not specific to the tail module, but rather Mediator *per se*, which is what we found. This may be due to overall transcription levels as suggested by Jeronimo, and we will now include this possibility, but the fact remains that SAGA genes are more affected.

It would be important to base the conclusions on PIC on a more direct measure of PIC assembly.

We completely agree with Reviewer 3 that our conclusions about Mediator not being a required component of the PIC should rely on a more direct measurement of the PIC. The revised paper now looks at TBP and TFIIB occupancy (point 6 of new experiments).